



**The impact of the South-East Madagascar bloom on the oceanic CO$_2$ sink.**

Nicolas Metzl[1], Claire Lo Monaco[1], Coraline Leseurre[1], Céline Ridame[1], Jonathan Fin[1],

Claude Mignon[1], Marion Gehlen[2], Thi Tuyet Trang Chau[2]

[1] Laboratoire LOCEAN/IPSL, Sorbonne Université-CNRS-IRD-MNHN, Paris, 75005, Fr

[2] Laboratoire LSCE/IPSL, CEA-CNRS-UVSQ, Université Paris-Saclay Gif-sur-Yvette, 91191, Fr

*Correspondence to*: Nicolas Metzl (nicolas.metzl@locean.ipsl.fr)

**Abstract**

We described new sea surface CO$_2$ observations in the southwestern Indian Ocean obtained in January 2020 when a strong bloom event occurred south-east of Madagascar and extended eastward in the oligotrophic Indian Ocean subtropical domain. Compared to previous years (1991-2019) we observed very low fCO$_2$ and dissolved inorganic carbon concentrations (C$_T$) in austral summer 2020, indicative of a biologically driven process. In the bloom the anomaly of fCO$_2$ and C$_T$ reached respectively -33 µatm and -42 µmol.kg$^{-1}$ whereas no change is observed for alkalinity (A$_T$). In January 2020 we estimated a local maximum of air-sea CO$_2$ flux at 27°S of -6.9 mmol.m$^{-2}$.d$^{-1}$ (ocean sink) and -4.3 mmol.m$^{-2}$.d$^{-1}$ when averaging the flux in the band 26-30°S. In the domain 25-30°S/50-60°E we estimated that the bloom led to a regional carbon uptake of about -1 TgC.month$^{-1}$ in January 2020 whereas this region was previously recognized as an ocean CO$_2$ source or near equilibrium during this season. Using a neural network approach that reconstructs the monthly fCO$_2$ fields we estimated that when the bloom was at peak in December 2019 the CO$_2$ sink reached -3.1 (±1.0) mmol.m$^{-2}$.d$^{-1}$ in the band 25-30°S, i.e. the model captured the impact of the bloom. Integrated in the domain restricted to 25-30°S/50-60°E the region was a CO$_2$ sink in December 2019 of -0.8 TgC.month$^{-1}$ compared to a CO$_2$ source of +0.12 (± 0.10) TgC.month$^{-1}$ in December when averaged over the period 1996-2018. Consequently in 2019 this region was a stronger CO$_2$ annual sink of -8.8 TgC.yr$^{-1}$ compared to -7.0 (±0.5) TgC.yr$^{-1}$ averaged over 1996-2018. In austral summer 2019/2020, the bloom was likely controlled by relatively deep mixed-layer depth during preceding winter (July-September 2019) that would supply macro and/or micro-nutrients as iron to the surface layer to promote the bloom that started in November 2019 in two large rings in the Madagascar Basin. Based on measurements in January 2020, we observed relatively high N$_2$ fixation rates (up to 18 nmol N.L$^{-1}$.d$^{-1}$) suggesting that diazotrophs could play a role on the bloom in the nutrient depleted waters. The bloom event in austral summer 2020, along with the new carbonate system observations, represents a benchmark case for complex biogeochemical model sensitivity studies (including N$_2$-fixation process and iron supplies) for a better understanding on the origin and termination of this still "mysterious" sporadic bloom and its impact on ocean carbon uptake in the future.

**1 Introduction**

In the south-western subtropical Indian Ocean a phytoplankton bloom, called the South-East Madagascar Bloom (SEMB) occurs sporadically during austral summer (December-March, Figure 1). Based on first years of SeaWIFS satellite Chlorophyll-a (Chl-a) observations in 1997-2001 the SEMB has been first recognized by Longhurst (2001) as the largest bloom in the subtropics, extending over 3000 x 1500 km in the Madagascar Basin. When the SEMB is well developed like in February-March 1999 (Longhurst, 2001), monthly mean Chl-a concentrations are higher than 0.5 mg.m$^{-3}$ within the bloom contrasting with the low Chl-a in the





surrounding oligotrophic waters (< 0.05 mg.m⁻³). For reasons still not fully understood, this bloom occurred in
specific years (1997, 1999 and 2000) but was absent or moderate during a strong El Niño - Southern Oscillation
(ENSO) event in 1998. Following the first study by Longhurst (2001), the frequency, extension, levels of Chl-a
concentration and processes that would control the SEMB and its variability have been investigated in several
studies (Srokosz et al, 2004; Uz, 2007; Wilson and Qiu 2008; Poulton et al 2009; Raj et al 2010; Huhn et al
2012; Srokosz and Quartly 2013). Most of these studies were based on Chl-a derived from remote sensing and
altimetry. They all concluded the need for *in-situ* observations to understand the initiation, extend and
termination of the SEMB. To our knowledge *in-situ* biogeochemical observations (Chl-a, phytoplanktonic
species and nutrients) within the SEMB region were only obtained during the MadEx experiment in February
2005 (Poulton et al 2009; Srokosz and Quartly 2013) a year when the bloom was not well developed (e.g. Uz,
2007; Wilson and Qiu 2008). The MadEx cruise was conducted above the Madagascar ridge and west of 51°E in
the Madagascar Basin. However, the eastward extension of the SEMB reached occasionally the central
oligotropic Indian subtropics (longitude 70°E, Figure 1b) where the bloom is transported and apparently
bounded by the South Indian Counter Current (SICC) around 25°S (Siedler et al 2006; Palastanga et al 2007;
Huhn et al 2012; Menezes et al 2014). Modelling studies also suggested an eastward propagation of the SEMB
through advection or eddy transport originating from the south-east coast of Madagascar (Lévy et al 2007;
Srokosz et al 2015; Dilmahamod, et al 2020) but a precise explanation of the internal (e.g. local upwelling,
Ekman pumping, meso-scale dynamics) or external processes (e.g. iron from rivers, coastal zones or sediments)
at the origin of this "mysterious" bloom is still missing.

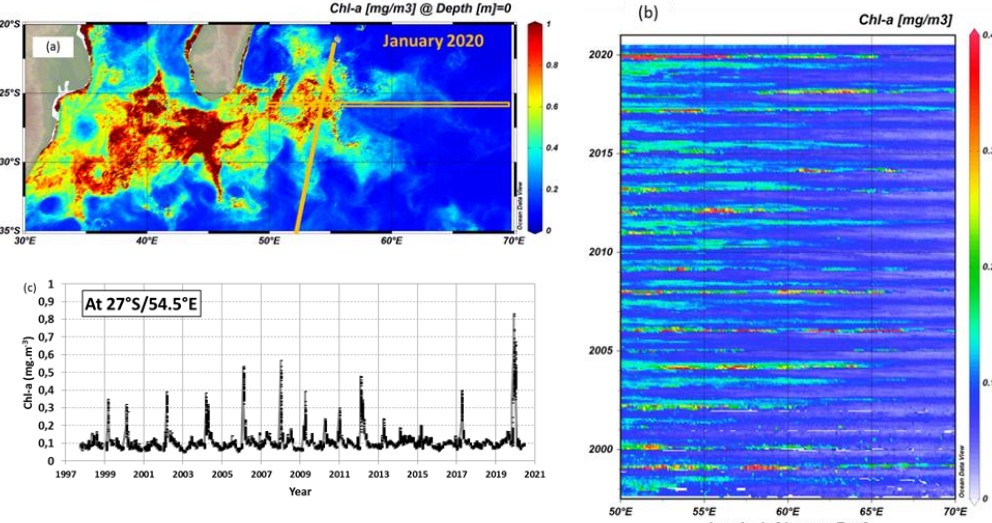

Figure 1: (a): Map of monthly surface Chl-a (mg.m⁻³) in the South-Western Indian Ocean in January 2020 derived from MODIS data (4x4km resolution), highlighting the bloom South and South-East of Madagascar. (b) Hovmoller Time-series (Time/Longitude) of Chl-a (mg.m⁻³) around 26.5°S along 50-70°E (Orange box in a). c) Time-series of monthly Chl-a (mg.m⁻³) in the box 27°S/54.5°E (only when valid number of pixels is greater than 5 for each point). The orange line on the map identifies the track of the OISO-30 cruise. The figures highlight the high Chl-a concentration in austral summer 2020. Figures (a) and (b) produced with ODV (Schlitzer, 2013) from data downloaded at https://resources.marine.copernicus.eu/ (OCEANCOLOUR_GLO_CHL_L4_REP_OBSERVATIONS_009_093), last access, 10-April-2021.



The above studies have been recently synthetized by Dilmahamod et al (2019) who also proposed an
index to determine the level of the SEMB (strong, moderate or absent) based on the difference in Chl-a
concentrations between the western and eastern regions centered respectively around 55°E and 80°E at 24-28°S.
Quoting Dilmahamod et al (2019): "The South-East Madagascar Bloom is one of the largest blooms in the
world. It can play a major role in the fishing industry, as well as capturing carbon dioxide from the atmosphere".
Although numerous cruises measuring sea surface $CO_2$ fugacity ($fCO_2$) were conducted since the nineties in the
south-western Indian Ocean region (Poisson et al., 1993; Metzl et al., 1995; Sabine et al 2000; Metzl, 2009), the
impact of the SEMB on air-sea $CO_2$ fluxes was not previously investigated. This is probably because the bloom
was not strong enough at the time of the cruises to identify large $fCO_2$ anomalies in this region. Therefore, the
temporal (seasonal and/or inter-annual) $fCO_2$ variability in the western and subtropical Indian Ocean is generally
interpreted by thermodynamics as the main control, biological activity and mixing processes being secondary
driving processes in this oligotrophic region (Louanchi et al, 1996; Metzl et al 1998; Sabine et al 2000;
Takahashi et al 2002). On the other hand, all climatologies based on observations suggest rather homogeneous
sea surface $fCO_2$ or dissolved inorganic carbon ($C_T$) fields in this region (Takahashi et al, 2002, 2009, 2014; Lee
et al, 2000; Sabine et al 2000; Bates et al 2006; Lauvset et al 2016; Zeng et al 2017; Broullón et al 2020; Keppler
et al 2020; Fay et al 2021; Gregor and Gruber 2021). This suggests that, although the SEMB and its extent have
been regularly observed since 1997 it seems to have a small effect on $fCO_2$ or $C_T$ spatial variations. However, in
austral summer 2019-2020, the SEMB was particularly pronounced reaching monthly mean Chl-a concentrations
up to 2.5 mg.m$^{-3}$ at the peak of the bloom in December 2019. It was clearly much stronger than previously
observed, at least since 1997 (Figure 1) and reflected in $fCO_2$ observations in this region (Figure 2).
In this analysis, we describe new oceanic carbonate system observations in surface waters obtained in
January 2020 associated to this very strong SEMB event and compare these observations with climatological
values and previous $fCO_2$ data when the SEMB was not well developed. We also evaluate the impact of the
bloom on air-sea $CO_2$ fluxes based on both observations and reconstructed monthly $fCO_2$ fields in the South-
Western Indian Ocean.


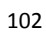

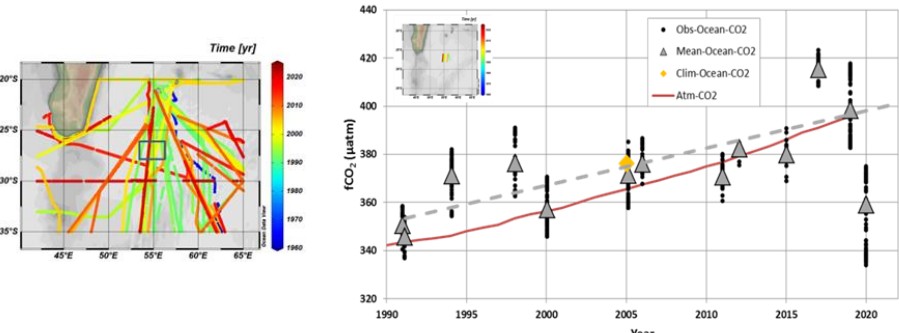



Figure 2: Left: Tracks of cruises with sea surface $fCO_2$ data available in the South-Western Indian Ocean in SOCAT data
product (version SOCAT-v2021, Bakker et al, 2016; 2021). Right: Time-series of $fCO_2$ data (black dots) and mean $fCO_2$ for
each period (grey triangles) in the box 27°S-28°S/55°E (black square in the map and insert on the right) for the months of
January and February (data available from 1991 to 2020 for austral summer). The red curve is the atmospheric $fCO_2$.
Although over 1991-2019 the ocean $fCO_2$ increased by +1.55 (± 0.40) µatm.yr$^{-1}$ (dashed grey line) due to anthropogenic $CO_2$
uptake, the $fCO_2$ recorded in January 2020 in the bloom were low compared to previous years with some values below 340
µatm, i.e. lower than in 1991. The January-February averaged $fCO_2$ in the same region derived from the 2005 climatology of
Takahasi et al (2014) is also plotted (orange diamond). Map on the left produced with ODV (Schlitzer, 2013).



## 2 Data collection

As part of the long-term OISO project (Ocean Indien Service d'Observations), the OISO-30 cruise was conducted in austral summer 2020 (from 2-January to 6-February 2020) onboard the R.V. Marion-Dufresne in the Southern Indian Ocean (part of the track shown in Figure 1). During the cruise, underway continuous surface measurements were obtained for temperature (SST), salinity (SSS), fugacity of $CO_2$ ($fCO_2$), total alkalinity ($A_T$) and total dissolved inorganic carbon ($C_T$). Analytical methods followed the protocol used since 1998 and previously described for other OISO cruises (e.g. Metzl et al 2006; Metzl, 2009; Lo Monaco et al, 2021). Sea surface temperature and salinity were measured continuously using a SBE45 thermosalinograph. Salinity data were controlled by regular sampling and conductivity measurements (Guildline Autosal 8400B and using IAPSO standard/OSIL). The SST and SSS data were also checked against CTD's surface records when available. Accuracies of SST and SSS are respectively 0.005 °C and 0.01. Total alkalinity ($A_T$) and total dissolved inorganic carbon ($C_T$) were measured continuously in surface water (3 to 4 sample/hour) using a potentiometric titration method (Edmond, 1970) in a closed cell. For calibration, we used the Certified Referenced Materials (CRMs, Batch #173) provided by Pr. A. Dickson (SIO, University of California). Replicate measurements were occasionally performed at the same location. At 30°S/54°E for 4 replicates the mean $A_T$ and $C_T$ concentrations were respectively 2328.6 (±0.7) and 1998.2 (±1.6) $\mu mol.kg^{-1}$. At 35°S/53.5°E for 6 replicates the mean $A_T$ and $C_T$ were 2340.5 (±0.6) and 2060.6 (±1.1) $\mu mol.kg^{-1}$. Overall, we estimated the accuracy for both $A_T$ and $C_T$ better than 3 $\mu mol.kg^{-1}$ (based on the analysis of CRMs). Like for all other OISO cruises, the surface underway $A_T$ and $C_T$ data will be available at NCEI/OCADS (www.ncei.noaa.gov/access/ocean-carbon-data-system/oceans/VOS_Program/OISO.html).

For $fCO_2$ measurements, sea-surface water was continuously equilibrated with a "thin film" type equilibrator thermostated with surface seawater (Poisson *et al.*, 1993). The $xCO_2$ in the dried gas was measured with a non-dispersive infrared analyser (NDIR, Siemens Ultramat 6F). Standard gases for calibration (271.39, 350.75 and 489.94 ppm) were measured every 6 hours. To correct $xCO_2$ dry measurements to $fCO_2$ *in situ* data, we used polynomials given by Weiss and Price (1980) for vapour pressure and by Copin-Montégut (1988, 1989) for temperature (temperature in the equilibrium cell measured using SBE38 was on average 0.28°C warmer than SST during the OISO-30 cruise). The oceanic $fCO_2$ data for this cruise are available in the SOCAT data product (version v2021, Bakker et al., 2016, 2021) and at NCEI/OCADS (Lo Monaco and Metzl, 2021). Note that when added to SOCAT, the original $fCO_2$ data are recomputed (Pfeil et al., 2013) using temperature correction from Takahashi et al (1993). Given the small difference between SST and equilibrium temperature, the $fCO_2$ data from our cruises are identical (within 1 $\mu atm$) in SOCAT and NCEI/OCADS. For coherence with other cruises we used the $fCO_2$ values as provided by SOCAT.

During the OISO-30 cruise, silicate (Si) concentrations in surface and water column samples (filtered at 0.2 $\mu m$, poisoned with 100 $\mu l$ $HgCl_2$ and stored at 5°C) were measured onshore by colorimetry (Aminot and Kérouel, 2007; Coverly et al. 2009). Based on replicate measurements for deep samples collected during OISO cruises we estimate an error of about 0.3 % in Si concentrations.

Unfiltered and 20$\mu m$-prefiltered seawater (~ 10m depth) were collected for the determination of net $N_2$ fixation in both the total fraction and the size-fraction lower than 20 $\mu m$ using the $^{15}N_2$ gas-tracer addition method (Montoya et al., 1996). By difference, we calculated $N_2$ fixation rates related to the microphytoplankton size class (> 20$\mu m$). Immediately after sampling, 2.5ml of 99% $^{15}N_2$ (Eurisotop) were introduced to 2.3L





polycarbonate bottles through a butyl septum. $^{15}N_2$ tracer was added to obtain a ~10% final enrichment. Then,
each bottle was vigorously shaken and incubated in an on-deck incubator with circulating seawater and equipped
with a blue filter to simulate the level of irradiance at the sampling depth. After 24h-incubation, 2.3L were
filtered onto pre-combusted 25mm GF/F filters, and filters were stored at −25°C. Sample filters were dried at
40°C for 48h before analysis. Nitrogen (N) content of particulate matter and its $^{15}N$ isotopic ratio were quantified
using an online continuous flow elemental analyzer (Flash 2000 HT), coupled with an Isotopic Ratio Mass
Spectrometer (Delta V Advantage via a conflow IV interface from Thermo Fischer Scientific). $N_2$ fixation rates
were calculated by isotope mass balanced as described by Montoya et al. (1996). The detection limit for $N_2$
fixation, calculated from significant enrichment and lowest particulate nitrogen is estimated to 0.04 nmol N $L^{-1}$ $d^-$
$^1$.
Other data used in this analysis (e.g. Chl-a from remote sensing, ADCP, current fields, $fCO_2$, $A_T$, $C_T$
from other cruises or from climatology) will be referred to in the next sections when appropriate.

**3 Reconstructed $fCO_2$ and air-sea $CO_2$ fluxes**

In order to complement the results based on regional *in-situ* data and evaluate the $CO_2$ sink anomalies in
this region back to 1996, we also used results from a neural network model that reconstructs monthly $fCO_2$ fields
and air-sea $CO_2$ fluxes. The $fCO_2$ fields were obtained from an ensemble-based feed-forward neural network
model (named CMEMS-LSCE-FFNNN) described in Chau et al (2021). To take into account the period in
austral summer 2020 when the SEMB was particularly strong, we used the latest temporal extension of the
model which relies on the most recent version of the SOCAT data-base (SOCAT-v2021, Bakker et al, 2021). For
a full description of the model, access to the data and a statistical evaluation of $fCO_2$ reconstructions please refer
to Chau et al (2021).

**4 Results**

**4.1 Sea surface $fCO_2$, $C_T$ and $A_T$ distributions in the SEMB in January 2020**

In January 2020, the SEMB occupied a large region in the Southern section of the Mozambique
Channel, the Natal Basin, the Mozambique Plateau and the Madagascar Basin. It extended eastward with meso-
scale and filaments structures reaching 60°E in the southern subtropical Indian Ocean where Chl-a was up to 0.5
mg.m$^{-3}$ (Figure 1a). Compared to previous years, the spatial structure of the 2020 SEMB event resembled to the
one that occurred in 2008 (e.g. Dilmahamod et al 2019), albeit with much higher Chl-a concentrations in 2020
(Figure 1b, c). As opposed to previous years, the 2020 SEMB event started in November 2019 in the Madagascar
Basin and was pronounced in two large rings with monthly mean Chl-a concentrations reaching 1 mg.m$^{-3}$ at
25°S/52°E (Supp Mat Figure S1). These large Chl-a rings were likely linked to eddies and/or to the retroflection
of the South-East Madagascar current, SEMC (Lutjeharms 1988; Longhurst 2001; de Ruijter et al 2004) as seen
in the surface currents fields in November 2019 (Supp Mat Figure S2). In December 2019, the surface of the
SEMB extended in all directions and a maximum monthly mean Chl-a concentration up to 2.9 mg.m$^{-3}$ was
detected around 25°S/51.5°E (Supp Mat Figure S1). The SEMB was less developed in late February 2020 (Supp
Mat Figure S1). Whatever the origin and multiple drivers of the SEMB in 2020 through internal or external
forcing (Dilmahamod et al 2019) this rather strong biological event would significantly drawdown the $C_T$
concentration and $fCO_2$ during several weeks from November 2019 to February 2020 in this region.



Along the OISO-30 cruise track at 54°E in January 2020, the underway surface measurements started at
26.5°S for $fCO_2$ and at 27°S for $A_T$ and $C_T$. Along this track the sea surface Chl-a concentrations were relatively
lower south of 27°S (0.2-0.4 mg.m$^{-3}$) than north of 27°S (0.8-1.2 mg.m$^{-3}$, Figure 3a). This was associated with a
rapid decrease in $fCO_2$ (Figure 3a) and salinity normalized $C_T$ (N-$C_T$ = $C_T$*35/SSS) concentration (Figure 3b).
Because there was a sharp gradient in salinity at that latitude (Supp Mat Fig S3), no significant change was
observed for salinity normalized $A_T$ (N-$A_T$ = $A_T$*35/SSS) along the track (Figure 3b).

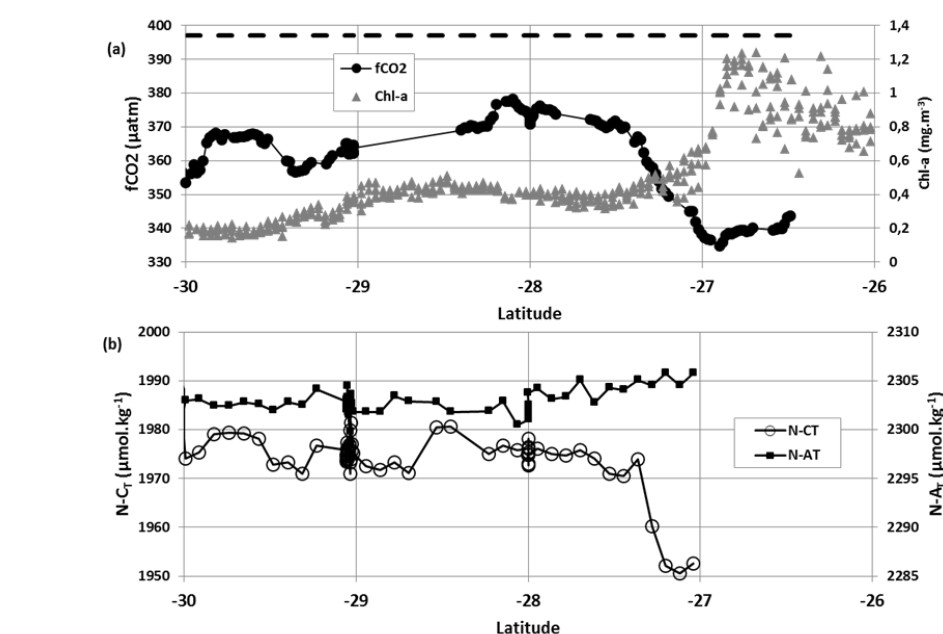

Figure 3: Top (a): Sea surface $fCO_2$ (µatm) measured in January 2020 (black circles) and Chl-a (mg.m$^{-3}$) from MODIS (4x4
km) along the cruise track (grey triangles). Bottom (b): Sea surface salinity normalized $C_T$ (N-$C_T$, open circles) and salinity
normalized $A_T$ (N-$A_T$, black squares) measured in January 2020 (both in µmol.kg$^{-1}$). Low $fCO_2$ and N-$C_T$ concentrations
recorded around 27°S were linked to high Chl-a (up to 1.2 mg.m$^{-3}$) in the SEMB. In (a) the dashed-line represents the average
atmospheric $fCO_2$ for January 2020.

The mean properties and differences within and out of the peak bloom are listed in Table 1. Although
the ocean was warmer in the bloom at 27°S (about +1°C, Supp Mat Fig. S3), $fCO_2$ was clearly much lower at
that location. The $fCO_2$ difference within and out of the peak bloom was -33 µatm based on $fCO_2$ measurements.
Given the error associated to the $fCO_2$ calculations using $A_T$ and $C_T$ data (±13 µatm, Orr et al 2018) the observed
$fCO_2$ difference is confirmed with $fCO_2$ calculated with the $A_T$-$C_T$ pairs (difference of -34.5 µatm, last column in
Table 1). If one takes into account the effect of the warming on $fCO_2$ (Takahashi et al, 1993), the $fCO_2$ in the
bloom would be 323.5 µatm. Therefore the solely impact of the biological processes in the bloom reduced $fCO_2$
by -49.3 µatm. This is a very large effect and coherent with the observed difference in N-$C_T$ of -23.4 µmol.kg$^{-1}$
within and out of the bloom and almost no change in N-$A_T$ (Table 1).





--------------------------------------------------------------------------------------------------------------------

Table 1: Mean properties and their difference observed in January 2020 within and out of the SEMB peak
bloom. For $fCO_2$, results based on measurements ($fCO_{2mes}$) or calculated using $A_T$-$C_T$ pairs ($fCO_{2cal}$) are both
listed. Standard deviations are indicated between brackets.

--------------------------------------------------------------------------------------------------------------------

| Region | SST °C | SSS PSU | Chl-a mg.m$^{-3}$ | $C_T$ µmol.kg$^{-1}$ | N-$C_T$ µmol.kg$^{-1}$ | $A_T$ µmol.kg$^{-1}$ | N-$A_T$ µmol.kg$^{-1}$ | $fCO_{2mes}$ µatm | $fCO_{2cal}$ µatm |
|---|---|---|---|---|---|---|---|---|---|
| Within Peak Bloom (Around 27°S) | 26.39 (0.21) | 35.22 (0.05) | 0.97 (0.18) | 1958.6 (2.5) | 1951.7 (1.0) | 2313.5 (2.7) | 2305.4 (0.7) | 339.5 (2.5) | 329.8 (2.0) |
| South of the Peak Bloom (Around 28°S) | 25.32 (0.10) | 35.48 (0.03) | 0.41 (0.04) | 2000.6 (2.2) | 1975.2 (1.4) | 2332.1 (1.9) | 2302.4 (1.3) | 372.8 (2.2) | 364.3 (2.6) |
| Difference In-Out | +1.07 | -0.26 | +0.56 | -42.0 | -23.4 | -18.6 | +3.0 | -33.3 | -34.5 |

--------------------------------------------------------------------------------------------------------------------

The atmospheric $xCO_2$ was 410 ppm in January 2020, equivalent to 397 µatm for $fCO_{2atm}$ (dashed line

in Figure 3a, where $xCO_2$ in ppm was corrected to $fCO_2$ according to Weiss and Price, 1980). Consequently the
region was a strong $CO_2$ sink within the bloom area with maximal $\Delta fCO_2$ value of -60 µatm at 27°S (where
$\Delta fCO_2 = fCO_{2oce} - fCO_{2atm}$). As a comparison at this location (28-24°S-52.5°E) the climatological $\Delta fCO_2$ value for
January (Takahashi et al 2009) was estimated between +4 to +10 µatm, i.e. a small source or near equilibrium. It
is well known that gas exchange at the air-sea interface depends on both $\Delta fCO_2$ and the wind speed (e.g.
Wanninkhof 2014). The net flux of $CO_2$ across the air-sea interface ($FCO_2$) was calculated according to the
following equation (1):

$FCO_2 = k\ K0\ \Delta fCO_2$                                         (Eq. 1)


Where $K0$ is the solubility of $CO_2$ in seawater calculated from *in situ* temperature and salinity (Weiss, 1974) and
$k$ (cm.h$^{-1}$) is the gas transfer velocity expressed from the wind speed U (m.s$^{-1}$) (Wanninkhof, 2014) and the
Schmidt number Sc (Wanninkhof, 1992) following equation (2):

$k = 0.251\ U^2\ (Sc/660)^{-0.5}$                              (Eq. 2)


In the region 25°S-30°S/45°E-60°E the average monthly wind speed (GMAO, 2015) was 7.9 m.s$^{-1}$ in

January 2020. This value is the same as derived from 6-hourly wind speed products at location 27°S-54°E, 7.8
(±2.3) m.s$^{-1}$ (Supp Mat Figure S4a). Using equation (1) and (2), this leads to a $CO_2$ sink of -6.9 mmol.m$^2$.d$^{-1}$ at
27°S in January 2020 whereas in the climatology (Takahashi et al 2009) this region was a $CO_2$ source of +0.72
mmol.m$^2$.d$^{-1}$ in January. In the band 26-30°S where Chl-a varied between 1.2 and 0.2 mg.m$^{-3}$ (Figure 3) the $CO_2$
sink was still significant on average, -4.3 (± 1.3) mmol.m$^2$.d$^{-1}$.

Integrated over 1 month and a surface of the bloom of 3000x1500 km (Longhurst, 2001), i.e. 4.5 Mkm$^2$,

the carbon uptake in January 2020 would be -7.2 (± 2.2) TgC.month$^{-1}$. However, based on the Chl-a distribution
in January 2020 (Figure 1a), we estimated the surface of the bloom east of 45°E to range between 1 and 1.7
Mkm$^2$ depending the criteria based on Chl-a concentrations (respectively Chl-a = 0.16 mg.m$^{-3}$ for a major bloom
or Chl-a = 0.07 mg.m$^{-3}$ for a bloom, Dilmahamod et al 2019). This leads to an integrated $CO_2$ sink ranging
between -1.7 and -2.7 TgC.month$^{-1}$ probably more realistic than when using the surface of the bloom as defined
by Longhurst (2001). When restricted to the surface of the domain 25-30°S/50-60°E (0.6 Mkm$^2$) the integrated
$CO_2$ sink in January 2020 based on $fCO_2$ observations would be -1.0 TgC.month$^{-1}$.



Given the $fCO_2$ distribution observed in January 2020 and the strong $CO_2$ sink evaluated within the
SEMB, we then compared the 2020 observations with a period when the bloom was absent (or small) and for
which $fCO_2$ data were also available for comparison.

**4.2 Comparison with a low bloom year: 2005**


For the period 1998-2016, Dilmahamod et al (2019) synthetized the season and years (their Table 1)
with strong or moderate SEMB and years when no bloom was clearly observed, such as in 2005. This is
confirmed from the Chl-a time series constructed around 27°S that showed low Chl-a in 2005 compared to 2004
and 2006 (Figure 1 b, c). However, it is worth to note that Poulton et al (2009) and Srokosz and Quartly (2013)
analyzed in-situ observations collected in this region in February 2005 during the MadEx cruise. They detected
that the bloom was present albeit with low Chl-a concentrations (maximum of 0.2 mg.m$^{-3}$). Based on surface
observations (Chl-a, species and nutrients) along a NE-SE transect between 47°E and 51°E, Srokosz and Quartly
(2013) reported that Chl-a variability around 50°E was strongly linked to eddy field as first noticed by Longhurst
(2001). They also observed from Seasoar fluorimeter data that the deep chlorophyll maximum (DCM) around
70-100m was relatively homogenous along the cruise track and not associated with eddy field as opposed to
surface Chl-a. Excepted for silicate that showed some low "patchy" concentrations (<1 µmol.kg$^{-1}$) associated
with filaments of higher Chl-a in the Madagascar Basin (Poulton et al, 2009), no significant variation was
observed for other nutrients during MadEx in February 2005 and this was probably the case for $fCO_2$.
Here we revisited the SEMB in austral summer 2005 using data collected during the OISO-12 cruise
(expocode 35MF20050113 in the SOCAT data product, Bakker et al, 2016). To compare with 2020, we selected
the $fCO_2$ data collected along the same track around 54°E in February 2005 (note that the $fCO_2$ data collected in
January 2005 to the east, around 60°E, were almost the same, not shown). In the region east of Madagascar, the
bloom was discernible around 25°S in January 2005 with maximum Chl-a concentrations around 0.3 mg.m$^{-3}$ at
50°E (Supp. Mat. Figure S5). In January, the bloom appeared to extend eastward following a large meandering
structure around 25°S and in February 2005 the bloom is even detectable at 65°E-70°E where Chl-a
concentration was on average 0.19 (± 0.03) mg.m$^{-3}$ within the core of the bloom. Interestingly this seems to be
centered in the core of the SICC (Huhn et al 2012) as revealed at 25°S by the ADCP observations obtained in
2005 along the OISO-12 cruise track as well as in surface current fields (Supp. Mat. Figure S6). Like in
November 2019 (Supp. Mat. Figure S2) there was a clear signal of the SEMC retroflection in January 2005 that
could explain the structure and eastward propagation of the bloom.
The bloom in 2005 was low (Srokosz and Quartly, 2013; Dilmahamod et al, 2019) and thus it had no
impact on the $fCO_2$ distribution. This is shown in Figure 4 were we compared $fCO_2$ observations along the same
track in February 2005 and January 2020. We present the results for $\Delta fCO_2$ along with sea surface Chl-a for each
period. In 2005 the sea surface $fCO_2$ was pretty homogeneous with values near the atmospheric $fCO_2$ level
($\Delta fCO_2$ close to 0). Although one would expect to observe higher $fCO_2$ 15 years later due to anthropogenic
carbon uptake by the ocean driven by the increase in atmospheric $CO_2$ (and thus about the same $\Delta fCO_2$), both
$fCO_2$ and $\Delta fCO_2$ in 2020 were much lower than in 2005 especially north of 27°S (Figure 4, Table 2). In austral
summer 2005, the region was near equilibrium with a $\Delta fCO_2$ mean value of +8.6 (± 7.1) µatm. This is close to
the climatology constructed for a reference year in 2005 (Takahashi et al, 2014, Table 2) and this is expected as
the climatology included the $fCO_2$ data from OISO cruises obtained in this region in 1998-2008. On the opposite,
in January 2020 we observed a strong sink (maximum $\Delta fCO_2$ = -60 µatm at 27°S). As the temperature was about





the same for both periods, the difference in $fCO_2$ was not due to thermodynamics and the $CO_2$ sink observed in
2020 was directly linked to the strong SEMB that occurred in austral summer.
The average monthly wind speed was also about the same in 2020 (7.9 m.s$^{-1}$) and 2005 (8.5 m.s$^{-1}$) (Supp
Mat. Fig S4b). Consequently the difference in the air-sea $CO_2$ flux between the two periods was controlled by
$\Delta fCO_2$. In the region 26-30°S/55°E, the mean $CO_2$ flux in 2005 was estimated at +1.2 mmol.m$^{-2}$.d$^{-1}$ (a source)
against -4.3 mmol.m$^{-2}$.d$^{-1}$ (a sink) in 2020.

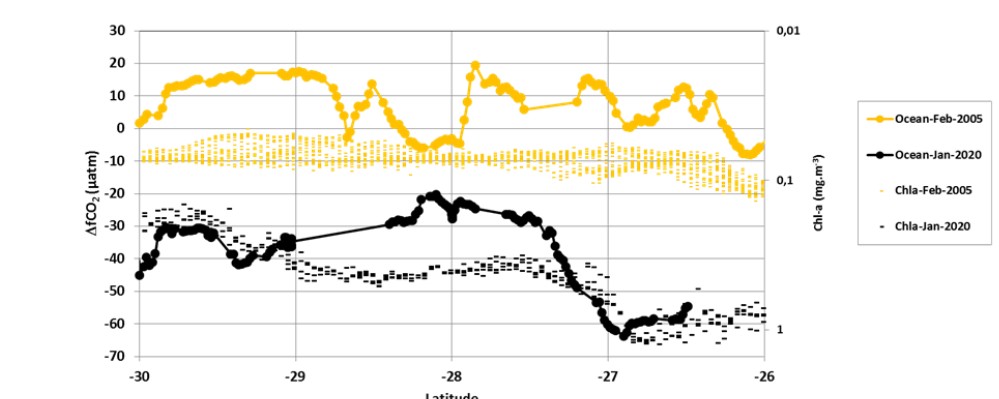

Figure 4: $\Delta fCO_2$ (µatm) ($\Delta fCO_{2\ =}$ $fCO2_{ocean}$-$fCO2_{atm}$) and sea surface Chl-a (mg.m$^{-3}$) distribution in January 2020 (black) and
February 2005 (orange) along the same track around 54°E in the South-Western Indian Ocean. Here Chl-a is in log10 scale
and inverted. In 2020 when the SEMB was particularly strong $\Delta fCO_2$ was negative (ocean $CO_2$ sink), whereas in 2005 when
the bloom was small, $\Delta fCO_2$ was close to 0 or positive (ocean $CO_2$ source).
Table 2: Mean sea surface properties observed along the same track in January 2020 and February 2005 in the
region 30°S-26°S/54°E. Also indicated the mean values in the same region and season from the climatology of
Takahashi et al (2014) and the Chl-a climatology evaluated for January-February 1998-2019. Nb is the number
of observations for SST, SSS and $fCO_2$. Standard deviations are indicated in bracket.

| Cruise | Period | SST (°C) | SSS (PSU) | $fCO_2$ (µatm) | $\Delta fCO_2$ (µatm) | Chl-a (mg.m$^{-3}$) |
|---|---|---|---|---|---|---|
| OISO-12 Nb= 115 | Feb-2005 | 25.443 (0.813) | 35.240 (0.112) | 374.2 (7.1) | +8.6 (7.1) | 0.087 (0.014) |
| OISO-30 Nb=217 | Jan-2020 | 25.103 (0.739) | 35.442 (0.110) | 362.2 (10.7) | -36.2 (10.7) | 0.489 (0.266) |
| Climatology | Jan-Feb | 26.242 (0.898) | 35.230 (0.140) | 376.1 (3.6) | +10.5 (3.6) | 0.105 (0.093) |

**5 Discussion**
**5.1 A large biologically driven $fCO_2$ negative anomaly in 2020 relative to the anthropogenic uptake of $CO_2$**
Like for $fCO_2$, the N-$C_T$ concentrations observed in the SEMB in January 2020 (1950 µmol.kg$^{-1}$, Figure
3b, Table 1) were low compared to the climatology (Takahashi et al 2014). At 24°S-28°S/54°E, the N-$C_T$
climatological value in January range between 1970 and 1980 µmol.kg$^{-1}$. As the climatology produced by





Takahashi et al (2014) was referred to a nominal year 2005, one would expect to observe higher N-$C_T$
concentrations in 2020 due to anthropogenic $CO_2$ uptake.

In the Indian Ocean the decadal change of anthropogenic CO2 ($C_{ant}$) was first evaluated by Peng et al

(1998) comparing data obtained in 1978 and 1995 north of 20°S. For the upper layer in the tropics (20°S-10°S)
Peng et al (1998) estimated an increasing rate of $C_{ant}$ of around 1.1 µmol.$kg^{-1}$.$yr^{-1}$. More recently, Murata et al
(2010) evaluated the changes of $C_{ant}$ concentrations between 1995 and 2003 in the South Indian Ocean
subtropics. They estimated a mean increase of $C_{ant}$ of +7.9 (± 1.1) µmol.$kg^{-1}$ over 8.5 years in the upper layers
that corresponds to a trend of +0.93 (± 0.13) µmol.$kg^{-1}$.$yr^{-1}$. In a global context, Gruber et al (2019 a, b)
estimated an accumulation of anthropogenic $CO_2$ ($C_{ant}$) of +14.3 (± 0.3) µmol.$kg^{-1}$ in surface waters of the south-
western Indian Ocean over 1994-2007, corresponding to an increasing rate in $C_{ant}$ of +1.10 (± 0.02) µmol.$kg^{-1}$.$yr^-$
$^1$. To confirm these $C_{ant}$ trends that were based on the $C_{ant}$ differences between two periods (1995-1978, 2003-
1995 or 2007-1994) we calculated the $C_{ant}$ concentrations and long-term trend using water-column data available
in 1978-2020 in the region 30-26°S/55°E. We extracted the data from the most recent GLODAP quality
controlled data product (version GLODAPv2-2021, Lauvset et al 2021a,b) completed with data from OISO
cruises in 2012-2018. To calculate $C_{ant}$ we used the TrOCA method developed by Touratier et al. (2007).
Because indirect methods are not suitable for evaluating $C_{ant}$ concentrations in surface waters due to gas
exchange and biological activity we selected the data in the layer 100-250m below the DCM. $C_{ant}$ concentrations
were calculated for each sample in that layer and then averaged for each period to estimate the trend (Figure 5).
As expected the $C_{ant}$ concentrations in subsurface increased significantly from 1978 to 2020 and the long-term
trend of +1.05 (± 0.08) µmol.$kg^{-1}$.$yr^{-1}$ over this period is close to previous estimates based on different periods
and approaches (Peng et al 1998; Murata et al, 2010; Gruber et al, 2019a).

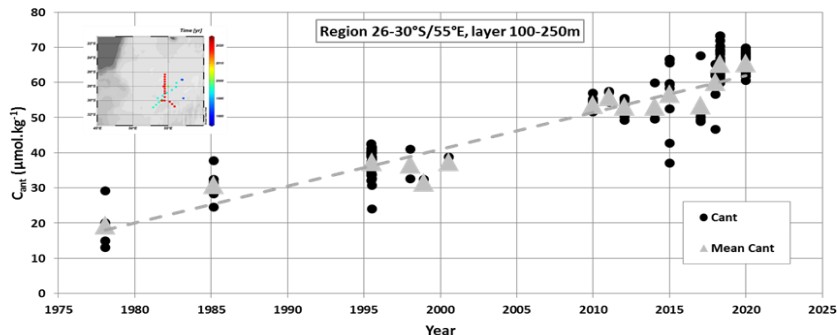

Figure 5: Time-series of anthropogenic $CO_2$ concentrations ($C_{ant}$) estimated in subsurface (layer 100-250m) in the region 26-
30°S/55°E from the GLODAPv2-2021 data product (Lauvset et al, 2021,a,b) completed with OISO cruises in 2012-2018
(location of selected stations in the insert map). The figure shows the $C_{ant}$ concentrations calculated for each sample (black
dots) and the $C_{ant}$ averaged in the layer 100-250m for each period (grey triangles). Over the period 1978-2020, the $C_{ant}$ long-
term trend is +1.05 (± 0.08) µmol.$kg^{-1}$.$yr^{-1}$ (dashed grey line).

Furthermore the $C_{ant}$ trend of around +1 µmol.$kg^{-1}$.$yr^{-1}$ is coherent with an increase in $C_T$ of between

+0.93 and +1.17 µmol.$kg^{-1}$.$yr^{-1}$ derived from the oceanic $fCO_2$ increase over the period 1991-2007 estimated
from winter and summer $fCO_2$ data (+1.75 and +2.2 µatm.$yr^{-1}$ respectively, Metzl, 2009) assuming constant
alkalinity and temperature. With the new data available after 2007, we have revisited the $fCO_2$ long-term trend
by selecting only the austral summer data in the region around 27°S-55°E (Figure 2). For the period 1991-2019



423 we estimated a fCO$_2$ trend of +1.55 (± 0.40) µatm.yr$^{-1}$. This is less than the atmospheric fCO$_2$ increase of +1.89

424 (± 0.03) µatm.yr$^{-1}$ over the same period suggesting that the CO$_2$ sink increased at this location. In a broader

425 context, Landschützer et al (2016) suggested that the carbon uptake tended to increase slightly in 1998-2011 in

426 the Subtropical Indian Ocean (their figure 3). We will see that such a change in the CO$_2$ fluxes in this region is

427 also revealed in the CMEMS-LSCE-FFNN model (Chau et al, 2021). Note that if at that location 27°S/55°E

428 (Figure 2) the ocean fCO$_2$ data in 2020 were also used to estimate the trend (1991-2020), the rate of fCO$_2$ would

429 be only +1.09 (± 0.48) µatm.yr$^{-1}$. i.e. about half the atmospheric fCO$_2$ trend. The fCO$_2$ observations in 2020

430 represent a large negative anomaly at local scale and thus caution is needed when incorporating such an anomaly

431 to detect and interpret long-term change in the CO$_2$ sink, at least in the south-western Subtropical Indian Ocean.

432   To compare the fCO$_2$ trends listed above with the anthropogenic rate of around +1.0 µmol.kg$^{-1}$.yr$^{-1}$

433 (Figure 5), we have calculated C$_T$ from the fCO$_2$ data and A$_T$ derived from salinity (described below). For this

434 calculation we used the CO2sys program (version CO2sys_v2.5, Orr et al., 2018) developed by Lewis and

435 Wallace (1998) and adapted by Pierrot et al. (2006) with K1 and K2 dissociation constants from Lueker et al.

436 (2000) and KSO$_4$ constant from Dickson (1990). The total boron concentration is calculated according to

437 Uppström (1974). For nutrients we fixed phosphate concentrations at 0 and silicate at 2.0 (± 0.6) µmol.kg$^{-1}$ (the

438 mean of 79 surface observations measured during previous OISO cruises in the region 22°S-30°S). To derive A$_T$

439 from salinity we used the surface A$_T$ observations obtained since 1998 in the subtropical south-western Indian

440 Ocean (OISO cruises). From these data we estimated a robust relationship (Figure 6):

442   A$_T$ (µmol.kg$^{-1}$) = 62.1601 * SSS + 123.1 (rms= 7.0 µmol.kg$^{-1}$, r= 0.89, n= 3400)   (Eq. 3)

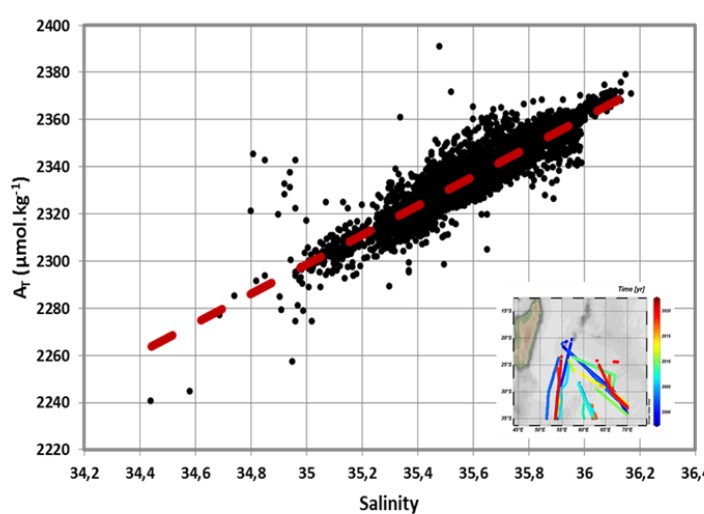

467 Figure 6: Relationship of A$_T$ (µmol.kg$^{-1}$) versus Salinity deduced from surface A$_T$ data (n= 3400) obtained during OISO
468 cruises in 1998-2020 in the South-Western Indian Ocean. For the subtropics we have selected the data in the region 35°S-
469 20°S/50°E-70°E (track of cruises shown in the insert map). The relationship (red dashed) is A$_T$ = 62.1601 * SSS +123.1 and
470 is used to calculate C$_T$ concentrations in this region (Figure 7). A$_T$ data are available at NCEI/OCADS
471 (https://www.ncei.noaa.gov/access/ocean-carbon-data-system/oceans/VOS_Program/OISO.html).

474   The use of other relationships (e.g. Millero et al 1998; Lee et al 2006) would change slightly the A$_T$

475 concentrations but not the interpretation on the C$_T$ trend in this region. The time-series of salinity normalized C$_T$

(N-$C_T$ = $C_T$*35/SSS) in the box 27°S-28°S/55°E shows that N-$C_T$ increased over the period 1991-2019 at a rate
of +0.70 (± 0.24) µmol.kg$^{-1}$.yr$^{-1}$ (Figure 7). This is somehow lower than the anthropogenic trend of +1 µmol.kg$^{-1}$
.yr$^{-1}$ suggesting that in addition to the anthropogenic $CO_2$ uptake, natural processes could also have a small
impact on the $C_T$ and f$CO_2$ trends in surface waters over almost 30 years.

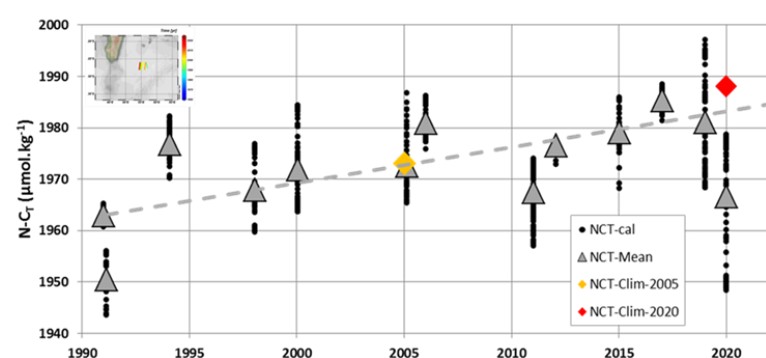

Figure 7: Time-series of salinity normalized $C_T$ (N-$C_T$ black dots) and their monthly mean (grey triangles) in the box 27°S-
28°S/55°E (insert map) calculated with f$CO_2$ observations (see figure 2) and reconstructed $A_T$ from salinity (Figure 6). The
figure shows data for the months of January and February (data available from 1991 to 2020 for austral summer). Over the
period 1991-2019, the N-$C_T$ trend is +0.70 (± 0.24) µmol.kg$^{-1}$.yr$^{-1}$ (dashed grey line) reflecting in part the anthropogenic $CO_2$
uptake. Note the low N-$C_T$ in January 2020 in the SEMB compared to previous years with some values around 1950
µmol.kg$^{-1}$ in 2020 as low as N-$C_T$ calculated in 1991. The N-$C_T$ concentration in the same region derived from the
climatology of Takahashi et al (2014) is also plotted (orange diamond for the reference year 2005) as well as the
climatological value for year 2020 after correcting for anthropogenic $CO_2$ (red diamond).

Having an estimate of the $C_T$ change due to anthropogenic $CO_2$ (around +1 µmol.kg$^{-1}$.yr$^{-1}$) and taking
into account this effect, the climatological N-$C_T$ concentration of 1973 µmol.kg$^{-1}$ for 2005 (Takahashi et al 2014)
corrected for the year 2020 would be 1988 µmol.kg$^{-1}$ in the region of interest. This is higher by up to +36
µmol.kg$^{-1}$ than the observed N-$C_T$ in January 2020 in the SEMB (Table 1, Figure 7). When correcting the
climatological value to the observed $C_T$ trend of +0.7 µmol.kg$^{-1}$.yr$^{-1}$, the N-$C_T$ in 2020 would be 1983.5 µmol.kg$^{-1}$
, i.e. +32.5 µmol.kg$^{-1}$ higher than the observed value in January 2020. The N-$C_T$ anomaly in January 2020 is
also large compared to the mean N-$C_T$ seasonal amplitude of 20 µmol.kg$^{-1}$ generally observed in the South
Indian subtropics (Metzl et al 1998; Takahashi et al 2014). We also note that climatological N-$A_T$ concentrations
of 2295 µmol.kg$^{-1}$ for January (Takahashi et al 2014) are very close to those we observed in January 2020 (Table
1, Figure 3b). Therefore the low f$CO_2$ and strong $CO_2$ sink in 2020 in the SEMB is due to a large drawdown of
$C_T$, i.e. not driven by temperature changes or alkalinity.

**5.2 Specificities of the SEMB bloom in 2020**

Based on previous studies it is likely that the biologically driven reduction of $C_T$ in the SEMB under
depleted sea surface nitrate concentrations was associated with the process of $N_2$ fixation (Uz, 2007). The
hypothesis that diazotrophy would play a role in the temporal $C_T$ (and thus f$CO_2$) variability is supported by the
observation of large $N_2$-fixing phytoplankton in the SEMB region in 2005 during MadEx cruise (Poulton et al
2009). These authors found that the filamentous cyanobacteria *Trichodesmium* was most abundant south of



Madagascar (over the Madagascar ridge) whereas diatom-diazotroph associations (as *Rhizosolenia/Richelia*)
were mainly observed east of Madagascar (in the Madagascar Basin).
Our measurements in January 2020 showed high spatial variability of the $N_2$ fixation rate (range from
0.8 to 18.3 nmol N.L$^{-1}$.d$^{-1}$, Figure 8). Such variability in the subtropical Indian ocean was also recently reported
by Hörstmann et al (2021) who measured $N_2$ fixation rates between 0.7 and 7.9 nmol N.L$^{-1}$.d$^{-1}$ in January-
February 2017 in the same region (OISO-27 cruise) but when the SEMB was not pronounced (Figure 1 b, c) and
when $fCO_2$ was high and above equilibrium (Figure 2). Our results for silicate (Si) and $N_2$-fix observations are
difficult to interpret because few samples were collected along the track (Figure 8). A maximum of $N_2$ fixation
rate was observed at 30°S that was not linked to changes in other properties. This local high $N_2$ fixation rate
could be related to *Trichodesmium* species but it was not sampled in January 2020. We also noted low Si
concentrations at 27°S (0.6 µmol.kg$^{-1}$) associated with higher Chl-a and lower $fCO_2$ and $C_T$ (Figure 3). The low
silicate might be associated with the presence of diatom-diazotroph associations (DDA) as observed during the
MadEx cruise (Poulton et al 2009). In the bloom $N_2$ fixation increased northward from 28°S (factor ~5). Based
on measurements for different size fractions we observed that the $N_2$ fixation is mainly related to the fraction >
20µm (i.e. Trichodesmium and DDA) representing 88% (± 9%) of the $N_2$ fixation. "Hotspots" of large
diazotrophs (20-180 and 180-2000 µm) were also detected in other regions of the south-western Indian Ocean in
May 2010 during the TARA expedition (Pierella Karlusich et al, 2021).

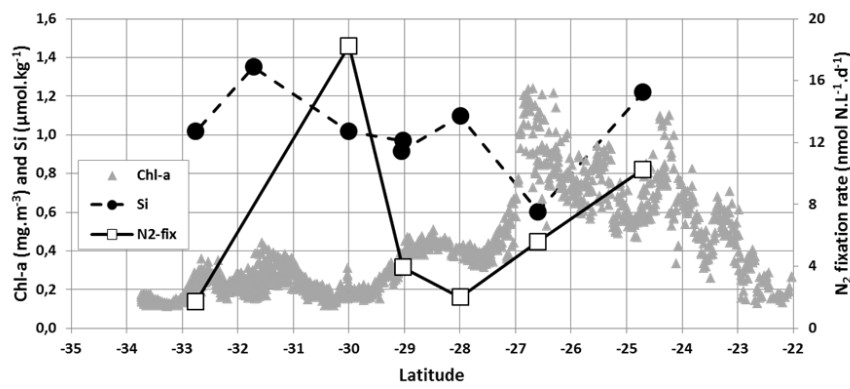

Figure 8: Sea surface silicate concentration (Si, µmol.kg$^{-1}$, black circles, scale on the left), $N_2$ fixation rate ($N_2$-fix, nmol N.L$^{-1}$
.d$^{-1}$, open squares, scale on the right) measured in January 2020 (OISO-30 cruise) and Chl-a (mg.m$^{-3}$, grey triangles, scale on
the left) from MODIS (4x4 km) along the cruise track. The low Si concentration (0.6 µmol.kg$^{-1}$) recorded around 27°S was
linked to higher Chl-a (up to 1.2 mg.m$^{-3}$) in the SEMB.

At global scale, the presence of $N_2$-fixers in the south-western Indian Ocean has been detected from
satellite data (Westberry and Siegel, 2006; Qi et al 2020) and relatively high $N_2$ fixation rates in austral summer
in this region were also derived from $N_2$-fix data using a machine learning approach (Tang and Cassar, 2019;
Tang et al, 2019). A large scale distribution of diazotrophy was further estimated from surface $C_T$ observations
suggesting the presence of $N_2$-fixers in the Mozambique Channel and the South-Western Indian Ocean (Lee et
al, 2002; Ko et al, 2018). These authors used regional N-$C_T$ versus SST relationships to reconstruct the N-$C_T$
field from which they estimated the net carbon production (NCP) in nitrate depleted waters, a proxy for carbon
production by $N_2$ fixing microorganisms. The N-$C_T$/SST relationship observed from in-situ data in January 2020
somehow mimics this process (Figure 9), i.e. the inter-annual variability of the N-$C_T$/SST relationship would
also inform on the NCP by $N_2$-fixers.



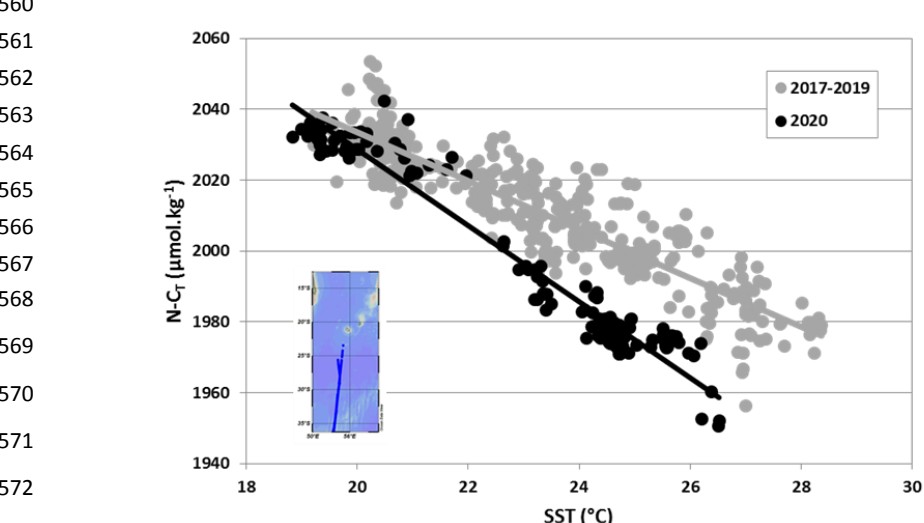

Figure 9: The relationship between N-$C_T$ ($\mu$mol.kg$^{-1}$) and SST in surface waters based on OISO cruises observations in the south-western Indian Ocean in austral summer 2017, 2018, 2019 and 2020 along the same repeated track (insert map). In January 2020 during the strong SEMB the N-$C_T$/SST relationship (black dots and black line) was much sharper than in 2017-2019 (grey dots and grey line) indicative of N$_2$-fix production in nitrate depleted waters (e.g. Ko et al 2018).

Sea surface warming and shallow mixed-layer depth (MLD) are proposed to lead to optimal conditions for the growth of the N$_2$-fixers and generate the SEMB (e.g. Longhurst, 2001; Srokosz et al 2015). In austral summer 2020, the ocean was not much warmer than previous years suggesting that temperature was not a specific driver of the SEMB that year. To the contrary, in January 2020 the region experienced a particularly shallow MLD which might have favored the bloom (observed MLD around 20m at 27°S-28°S, Supp. Mat. Figures S7 and S8).

As noted above, the strong bloom started in November 2019 and could be well identified in two large rings (Supp. Mat. Figure S1). In the northern ring at 25°S-52°E the MLD was deep (> 80m) during 3 consecutive months in July-September 2019 and deeper compared to previous years (Supp. Mat. Figure S9). This would have injected nutrients (and maybe iron) in surface layers and when the MLD was shallow at that location (< 20 m) the bloom developed in November 2019 and reached high Chl-a in December 2019 (up to 1.8 mg.m$^{-3}$). As the bloom covered a large region in December 2019 and January 2020 other specific processes like iron supply (from dust, coastal zone, rivers or sediments) still need to be identified to fully explain 2020 SEMB dynamics. The 2020 bloom was clearly recognized in Chl-a, fCO$_2$ and C$_T$ observations but at that stage we have no clear explanation on the process (or multiple drivers) that generated its extend and intensity.

**5.3 The changing ocean CO$_2$ uptake in the SEMB based on reconstructed pCO2**

The results presented above were based on local underway fCO$_2$ observations and the integrated air-sea CO$_2$ fluxes were thus extrapolated from local data on a surface representing the area covered by the bloom leading to a carbon uptake of between -1.7 and -2.7 TgC.month$^{-1}$ in January 2020. In the domain 25-30°S/50-60°E we estimated a CO$_2$ sink in January 2020 close to -1 TgC.month$^{-1}$.





To evaluate the impact of the bloom at the regional scale, we used monthly surface ocean $pCO_2$ and air-
sea $CO_2$ flux fields reconstructed by a neural network method as described in section 3 (CMEMS-LSCE-FFNN,
Chau et al, 2021). The SEMB was well developed in December 2019 and we can evaluate its impact on the air-
sea $CO_2$ fluxes by comparing December 2018 (low bloom) and December 2019 (strong bloom, Figure 10).

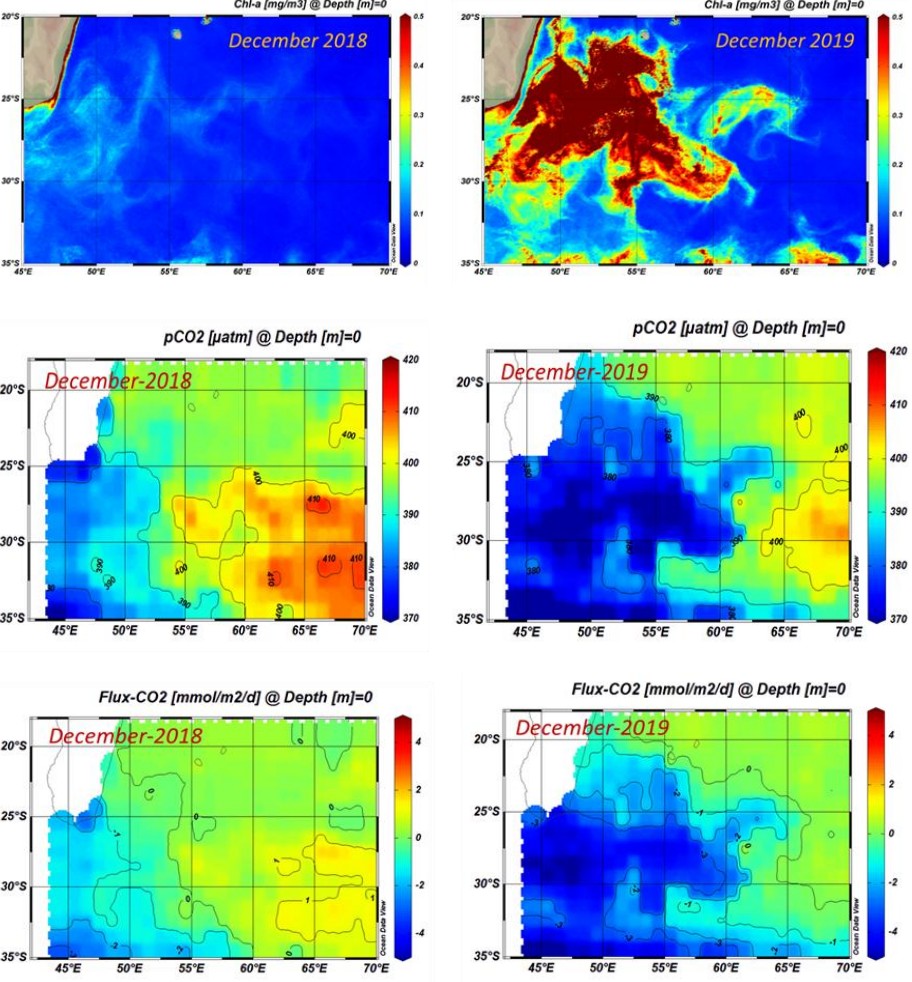

Figure 10: Maps of Chl-a (mg.m$^{-3}$), pCO2 (µatm) and the air-sea $CO_2$ fluxes (mmol.m$^{-2}$.d$^{-1}$) in the South-Western Indian
Ocean in December 2018 (left) and December 2019 (right). In December 2019 when the SEMB was particularly strong, the
$pCO_2$ was lower and air-sea $CO_2$ fluxes were negative (ocean sink, in blue), whereas in December 2018 when the bloom was
small, the fluxes were near equilibrium or positive in this region (ocean source, green-red). Chl-a data downloaded at
https://resources.marine.copernicus.eu/ (OCEANCOLOUR_GLO_CHL_L4_REP_OBSERVATIONS_009_093), last access,
10-April-2021. Figures produced with ODV (Schlitzer, 2013)

In the region 25-30°S/50-60°E, the average $pCO_2$ in December 2019 (375.9 ±6.3 µatm) was much lower
than in December 2018 (396.6 ±6.0 µatm) and thus opposite of the expected $pCO_2$ increase due to anthropogenic
$CO_2$ uptake. At the local scale, within the bloom at 27°S/54°E or at 29°S/50°E the CMEMS-LSCE-FFNN model
estimated low $pCO_2$ clearly linked to higher Chl-a in December 2019 (Supp. Figures S10, S11). Consequently



the region was a small $CO_2$ source of +0.07 (± 0.53) mmol.m$^{-2}$.d$^{-1}$ in December 2018 but a $CO_2$ sink in
December 2019 of -3.1 (± 1.0) mmol.m$^{-2}$.d$^{-1}$. Integrated over the region 25-30°S/50-60°E the carbon uptake
changed from a small $CO_2$ source in December 2018 of +0.019 TgC.month$^{-1}$ to a $CO_2$ sink in December 2019 of
-0.8 TgC.month$^{-1}$ (Supp Mat Figure S12) close to the estimate derived from observations in January 2020 (-1.0
TgC.month$^{-1}$). Over the period 1996-2018, the model evaluates each year a $CO_2$ source in December averaging
+0.12 (± 0.10) TgC.month$^{-1}$. This suggests that in late 2019 the CMEMS-LSCE-FFNN model did capture the
effect of the SEMB on pCO$_2$ and $CO_2$ fluxes, leading to a stronger regional $CO_2$ annual sink in 2019 (-8.8
TgC.yr$^{-1}$) compared to previous years (Figure 11).

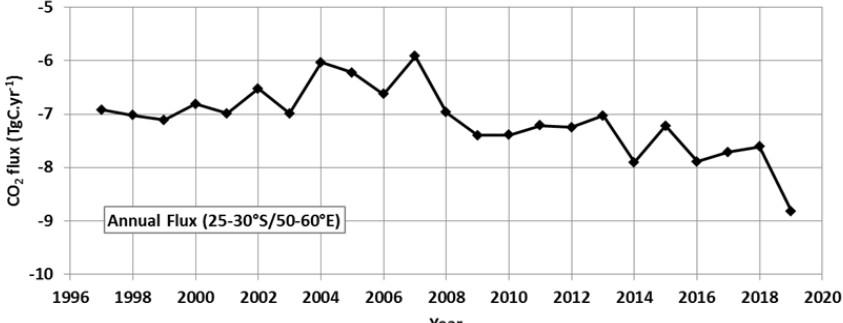

Figure 11: Annual air-sea $CO_2$ flux (TgC.yr$^{-1}$) in the South-Western Indian Ocean (region 25-30°S/50-60°E) for the period
1996-2019 from the CMEMS-LSCE-FFNN model. The carbon uptake progressively increased after 2007 with a maximum
$CO_2$ sink estimated in 2019 when the SEMB was particularly strong.

**6. Conclusions**

The new observations in the South-Western Indian Ocean presented here showed that the fCO$_2$ and $C_T$

concentrations in January 2020 were very low and far from normal conditions since 1991. This is explained by
the strong SEMB event that started in November 2019 in this region and was well developed in December 2019
and January 2020. Thanks to the continuous ocean color satellite data since 1997, the time-series of Chl-a in this
region showed that the bloom was particularly strong in austral summer 2019/2020. We suspect that prior to
1997, the SEMB has been less intense as suggested by *in-situ* fCO$_2$ data in 1991-1994 (Figure 2). We estimated
that the SEMB led to a regional carbon uptake of between -1.7 and –2.7 TgC.month$^{-1}$ in January 2020. The
variation of the regional ocean $CO_2$ sink due to the SEMB developed in late 2019 was also quantified with the
CMEMS-LSCE-FFNN model. Model results indicate a large anomaly in December 2019 that led to an annual
sink of -8.8 TgC.yr$^{-1}$, i.e. about 1 TgC.yr$^{-1}$ larger than previous years. The strong bloom in austral summer 2020
represents an interesting benchmark case to test models for a better understanding of the origin of the SEMB and
its impact on the regional ocean $CO_2$ sink. Future studies should target sensitivity analysis with complex
biogeochemical models including the $CO_2$ system, at different spatial resolution for the dynamics, and with (or
without) N$_2$ fixers (e.g. Monteiro et al 2010; Landolfi et al 2015; Paulsen et al 2017). This plankton functional
type is not yet included to models dedicated to this region (Srokosz et al 2015, Dilmahamod et al 2020). The new
fCO$_2$, $C_T$, $A_T$ and N$_2$ fixation rate observations presented here along with historical data (e.g. SOCAT, Bakker et
al 2016, 2021, Figure 2) could serve as a validation to compare periods with or without bloom. In the future, if
the SEMB as observed in 2020 is more frequent or becomes a regular situation and if organic matter is exported





below the surface mixed layer, this could represent a negative feedback to the ocean carbon cycle, i.e. the ocean
sink would be enhanced. As already noted by several authors (e.g. Dilmahamod et al 2019) dedicated studies in
this region, including the sampling of plankton, nutrients (e. g. iron), but also the determination of rates (e.g. $N_2$-
fixation) etc… would be relevant to understand the processes controlling the SEMB and to evaluate its impact on
the biological carbon pump.

**Data availability**

Data used in this study are available in SOCAT (www.socat.info) for $fCO_2$ surface data, in GLODAP
(www.glodap.info) for water-column data, at NCEI/OCADS (www.ncei.noaa.gov/access/ocean-carbon-data-
system/oceans/VOS_Program/OISO.html) for $A_T$-$C_T$ surface data, at Jas-ADCP
(http://uhslc.soest.hawaii.edu/sadcp) for ADCP data. The CMEMS-LSCE-FFNN model data are available at
E.U. Copernicus Marine Service Information (https://resources.marine.copernicus.eu/products).

**Authors contributions**

CLM and NM are co-Is of the ongoing OISO project. $fCO_2$, $A_T$ and $C_T$ data for OISO-30 were measured by
CLM, CL and CM and qualified by CLM and NM. Nutrients data for OIS0-30 were measured and qualified by
CL. $N_2$-fix data for OIS0-30 were measured and qualified by CR. CLM, NM, and JF qualified $fCO_2$, $A_T$ and $C_T$
data for previous OISO cruises. MG and TTTC developed the CMEMS-LSCE-FFNN model and provided the
model results. NM started the analysis, wrote the draft of the manuscript and prepared the figures with
contributions from all authors.

**Competing interest**

The authors declare that they have no conflict of interest.

**Acknowledgments**

The OISO program was supported by the French institutes INSU (Institut National des Sciences de l'Univers)
and IPEV (Institut Polaire Paul-Emile Victor), OSU Ecce-Terra (at Sorbonne Université), and the French
program SOERE/Great-Gases. We thank the French oceanographic fleet ("Flotte océanographique française")
for financial and logistic support to the OISO program and the OISO-30 oceanographic campaign
(https://doi.org/10.17600/18000679). We thank the captains and crew of *R.R.V. Marion Dufresne* and the staff at
IFREMER, GENAVIR and IPEV. $N_2$ fixation analysis was also supported by the French Research Program
LEFE (Les Enveloppes Fluides et l'Environnement) through ITALIANO project and we thank Magloire
Mandeng-Yogo and Fethiye Cetin for the measurements performed at the ALYSES plate-form (OSU Ecce-
Terra). The development of the neural network model benefited from funding by the French INSU-GMMC
project "PPR-Green-Grog (grant no 5-DS-PPR-GGREOG), the EU H2020 project AtlantOS (grant no 633211),
as well as through the Copernicus Marine Environment Monitoring Service (project 83-CMEMS-TAC-MOB).
The Surface Ocean $CO_2$ Atlas (SOCAT, www.socat.info) is an international effort, endorsed by the International
Ocean Carbon Coordination Project (IOCCP), the Surface Ocean Lower Atmosphere Study (SOLAS) and the
Integrated Marine Biogeochemistry and Ecosystem Research program (IMBER), to deliver a uniformly quality-
controlled surface ocean CO2 database.





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
