# Peer review of "The impact of the South-East Madagascar bloom on the oceanic CO2 sink."

_Biogeosciences, 2021_

## Author Comment (AC1)

Response to reviews of the paper: "The impact of the South-East Madagascar bloom on the oceanic CO2 sink" by Nicolas Metzl et al., Biogeosciences Discuss., https://doi.org/10.5194/bg-2021-283-RC1, 2021.

Reviewer 1, Meric Srokosz

Reviewer comments in italic, our responses in red

We thanks reviewer 1, Meric Srokosz, for his rapid and enthusiastic review.

*General Comment:*

*This is an interesting paper on the impact of the Southeat Madagascar Bloom on the oceanic CO2 sink, a problem that has not been previously addressed (to my knowledge). The results are convincing but critically depend on the work of Chau et al. (2021) which is in review. As my expertise is more on the impact of physical processes on the bloom biology I am unable to to make a judgment on the work of Chau et al. (2021) regarding the reconstruction of surface pCO2 and air-sea CO2 fluxes on which the results in this paper so much depend. Assuming that Chau et al. (2021) is correct / passes the review process and is published, I think that application of that approach to the Madgascar Bloom provides some new insights into its role as a CO2 sink. The observational aspects of the paper clearly show the strong interannual variability associated with the bloom, but the interpretation of these in terms of CO2 hinges on Chau et al. (2021). In light of this I am happy to recommend publication once the Chau et al. (2021) has passed peer review and been accepted for publication, after correction of a few minor issues (listed below).*

The reviewer commented (like reviewer 2) on the fact the results critically depend on the work of Chau et al. (2021), a paper that was in review at the time we submitted our paper. We agree that the model is an important piece of the analysis as it specifically offered an estimate of the impact of the bloom on $CO_2$ fluxes over a large domain (not only along the repeated cruise tracks) and also on an annual scale (not only for one season).

The paper submitted by Chau et al (2021) was not reviewed at the time we prepared our paper. However, the reviews for their paper were posted on-line in late September and the authors responded to the reviews last November 15th. At the time of this response (3/1/2022), the paper by Chau et al (2021) is not yet accepted but a revision is on the way.

Note that when we started this analysis (last year) we explored a first version of the same model described by Denvil-Sommer et al (2019) and Chau et la (2020) and preliminary results motivated the use of such a model for our purpose. The results of the first version of the model were also successfully used in another study focused in the Mozambique Channel (Lo Monaco et al, DSR 2021) and we were confident with this approach to explore the results of the model in the South-Western Indian sector. The model developed by Chau et al (2020, 2021) is an improvement of the first version by Denvil-Sommer et al (2019). Model results are under quality control of the European Copernicus Marine Environment Monitoring Service (CMEMS) and available for public use since 2019 (https://resources.marine.copernicus.eu/?option=com_csw&view=details&product_id=MULTIOBS_GLO_BIO_CARBON_SURFACE_REP_015_008, product DOI: https://doi.org/10.48670/moi-00047).

As the model has been updated, we thought it was more appropriate to use the new version of this model (Chau et al, 2021) extended to end-2019 that included the bloom anomaly well reproduced for December 2019 (Figures 10, 11).

Note also that results of the model by Chau et al (2021) have been used for global estimate of the ocean $CO_2$ sink in the last Global Carbon Budget (Friedlingstein et al, 2021; T. Chau is coauthor and results presented at COP26 last November). Similarly, results of the CMEMS-LSCE-FFNN model (Denvil-Sommer et al 2019, Chau et al 2020) were also used in the previous GCB (Friedlingstein et al, 2019, 2020). Here we show the model is relevant for regional analysis, i.e. not only a view of the global ocean carbon sink as used in the Global Carbon Project. We hope that by the time our present paper is revised, the paper by Chau et al (2021) will be accepted.

*Minor comments:*

*lines 275-277 - given that wind speed is spatially (as well as temporally) variable I am not sure how taking account of spatial variability would affect the calculations described here. Does it matter?*

Response: This is an important point, not only for air-sea $CO_2$ fluxes but also regarding the impact of the wind speed on mixing and circulation in this region and thus on biogeochemical properties (e.g. nutrients input) and $fCO_2$ distributions. To compare the fluxes between various periods (e.g. 2005 versus 2020) we looked at the wind speed for these periods; the averaged wind speed was almost the same for both periods (Supp Figure S4) and we concluded that the wind was not impacting the differences of fluxes that are mainly controlled by $\Delta fCO_2$ (2020 versus 2005, Table 2 or Figure 4). This is also clearly revealed in the model results when comparing December 2018 and 2019 (Figure 10). Note that when using only the $fCO_2$ data from the cruises, uncertainty of the flux integrated over a large domain (not only along the cruise track) is mainly linked to the prescribed "size" of the bloom (as discussed lines 281-228). This is why we also listed the results for different areas representative of the bloom. Finally, as the wind speed is around 8 $m.s^{-1}$ on average, the use of a different gas transfer coefficient (the term "k" in Equations 1, 2) would not introduce a bias when comparing the fluxes for different periods as "k" is almost the same at that wind speed depending the relations used (e.g. Wanninkhof, 2014).

*Figure 3 mark positions of eddies*

Response: The reviewer suggests to mark the positions of eddies in Figure 3 and S3. The aim of these figures was to present the records of properties ($fCO_2$, $C_T$, $A_T$, SST, SSS) along the cruise track in January 2020 and to highlight the contrast of $fCO_2$ and $C_T$ concentrations in the bloom (at 27°S) and south of the bloom as identified from Chl-a concentrations along the track also shown in Figure 3. As opposed to the Chl-a distribution in November 2019 (2 larges Chl-a rings probably associated to Eddies, Supp Figure S1 and S2 and discussed lines 189-193 and 585-593), the Chl-a in January 2020 covered a large domain, not clearly associated to eddies (Figures 1a and S1). The gradient of properties $fCO_2$, $C_T$ and salinity in January 2020 around 54°E (Figures 3 and S3) were clearly most pronounced around 27°S, whereas temperature did not show a specific signal (SST decreased progressively southward, Figure S3). On the other hand, salinity suggests the presence of multiple fronts localized around 26.5°S, 27°S and 28°S (Figure S3). Are these structures observed in January 2020 along the cruise track related to eddies or large scale circulation ?

Based on ADCP data recorded in January 2020 (See figure R1 in this response, a figure that was not included in the submitted manuscript) the structure of the current appeared mainly zonal with marked East/West components (i.e. not clearly associated to Eddy). The ADCP section shows a relatively strong westward current around 27°S, down to 600m. Interestingly, it seems that the SICC well observed in 2005 around 25°S (Figure S6) was not so well marked in January 2020 (with a subsurface current centered around 200m at 25°S).

We also plot the $fCO_2$ and salinity records (like original Figures 3 and S3) here along with surface current speed (figure R2 in this response). The low salinity and low $fCO_2$ around 27°S (in the bloom) were associated with the westward current; in the south, when salinity and $fCO_2$ were higher the current was also westward suggesting this is not a signal of Eddy.

To conclude, we were not able to detect Eddies along the track in January 2020 and thus we did not add any marks in Figures 3 and S3 as suggested by the reviewer. The question of the link between eddies, $fCO_2$ distribution and associated air-sea $CO_2$ flux is a relevant topic but we think it is beyond the scope of our analysis mainly focusing on the large scale (i.e. comparing data in a large domain over the period 1991-2020). Unfortunately we have no in-situ data in November 2019 that would cross the Eddy-like structures of the bloom (Figure S1, S2) as was observed during the MadEX cruise in February 2005 (Poulton et al, 2009; Srokosz and Quartly, 2013, discussed lines 298-307). On this issue (the link between eddies and bloom) we will add a few words to the conclusion to suggest this is also a relevant research topic for future in-situ studies especially in this highly complex dynamical region. Note for the editor, that concerning this specific question we contacted the reviewer to be sure that his question concerned Figure 3 and no other figures.

*Figure S2 add lat & long on axes, also Figure S6 bottom:*
Response: Latitude and longitude will be added in the maps. In Figure S2 we will also add two large circles to highlight the locations of the bloom as identified in Figure S1 for November 2019.

:::::::::::::::::
;;;;;;;;;;;;;;;;;

References added in this response (not listed in the manuscript):

Chau, T. T., Gehlen, M., and Chevallier, F., 2020: Global Ocean Surface Carbon Product MULTIOBS_GLO_BIO_CARBON_SURFACE_REP_015_008, E.U. Copernicus Marine Service Information. "https://resources.marine.copernicus.eu/documents/QUID/CMEMS-MOB-QUID-015-008.pdf".

Denvil-Sommer, A., Gehlen, M., Vrac, M., and Mejia, C.: LSCE-FFNN-v1: a two-step neural network model for the reconstruction of surface ocean $p$CO$_2$ over the global ocean, Geosci. Model Dev., 12, 2091-2105, https://doi.org/10.5194/gmd-12-2091-2019,2019.

Friedlingstein, P., Jones, M. W., O'Sullivan, M., Andrew, R. M., Hauck, J., Peters, G. P., Peters, W., Pongratz, J., Sitch, S., Le Quéré, C., Bakker, D. C. E., Canadell, J. G., Ciais, P., Jackson, R. B., Anthoni, P., Barbero, L., Bastos, A., Bastrikov, V., Becker, M., Bopp, L., Buitenhuis, E., Chandra, N., Chevallier, F., Chini, L. P., Currie, K. I., Feely, R. A., Gehlen, M., Gilfillan, D., Gkritzalis, T., Goll, D. S., Gruber, N., Gutekunst, S., Harris, I., Haverd, V., Houghton, R. A., Hurtt, G., Ilyina, T., Jain, A. K., Joetzjer, E., Kaplan, J. O., Kato, E., Klein Goldewijk, K., Korsbakken, J. I., Landschützer, P., Lauvset, S. K., Lefèvre, N., Lenton, A., Lienert, S., Lombardozzi, D., Marland, G., McGuire, P. C., Melton, J. R., Metzl, N., Munro, D. R., Nabel, J. E. M. S., Nakaoka, S.-I., Neill, C., Omar, A. M., Ono, T., Peregon, A., Pierrot, D., Poulter, B., Rehder, G., Resplandy, L., Robertson, E., Rödenbeck, C., Séférian, R., Schwinger, J., Smith, N., Tans, P. P., Tian, H., Tilbrook, B., Tubiello, F. N., van der Werf, G. R., Wiltshire, A. J., and Zaehle, S.: Global Carbon Budget 2019, Earth Syst. Sci. Data, 11, 1783–1838, https://doi.org/10.5194/essd-11-1783-2019, 2019.

Friedlingstein, P., O'Sullivan, M., Jones, M. W., Andrew, R. M., Hauck, J., Olsen, A., Peters, G. P., Peters, W., Pongratz, J., Sitch, S., Le Quéré, C., Canadell, J. G., Ciais, P., Jackson, R. B., Alin, S., Aragão, L. E. O. C., Arneth, A., Arora, V., Bates, N. R., Becker, M., Benoit-Cattin, A., Bittig, H. C., Bopp, L., Bultan, S., Chandra, N., Chevallier, F., Chini, L. P., Evans, W., Florentie, L., Forster, P. M., Gasser, T., Gehlen, M., Gilfillan, D., Gkritzalis, T., Gregor, L., Gruber, N., Harris, I., Hartung, K., Haverd, V., Houghton, R. A., Ilyina, T., Jain, A. K., Joetzjer, E., Kadono, K., Kato, E., Kitidis, V., Korsbakken, J. I., Landschützer, P., Lefèvre, N., Lenton, A., Lienert, S., Liu, Z., Lombardozzi, D., Marland, G., Metzl, N., Munro, D. R., Nabel, J. E. M. S., Nakaoka, S.-I., Niwa, Y., O'Brien, K., Ono, T., Palmer, P. I., Pierrot, D., Poulter, B., Resplandy, L., Robertson, E., Rödenbeck, C., Schwinger, J., Séférian, R., Skjelvan, I., Smith, A. J. P., Sutton, A. J., Tanhua, T., Tans, P. P., Tian, H., Tilbrook, B., van der Werf, G., Vuichard, N.,

Walker, A. P., Wanninkhof, R., Watson, A. J., Willis, D., Wiltshire, A. J., Yuan, W., Yue, X., and Zaehle, S.: Global Carbon Budget 2020, Earth Syst. Sci. Data, 12, 3269–3340, https://doi.org/10.5194/essd-12-3269-2020, 2020.

Friedlingstein, P., Jones, M. W., O'Sullivan, M., Andrew, R. M., Bakker, D. C. E., Hauck, J., Le Quéré, C., Peters, G. P., Peters, W., Pongratz, J., Sitch, S., Canadell, J. G., Ciais, P., Jackson, R. B., Alin, S. R., Anthoni, P., Bates, N. R., Becker, M., Bellouin, N., Bopp, L., Chau, T. T. T., Chevallier, F., Chini, L. P., Cronin, M., Currie, K. I., Decharme, B., Djeutchouang, L., Dou, X., Evans, W., Feely, R. A., Feng, L., Gasser, T., Gilfillan, D., Gkritzalis, T., Grassi, G., Gregor, L., Gruber, N., Gürses, Ö., Harris, I., Houghton, R. A., Hurtt, G. C., Iida, Y., Ilyina, T., Luijkx, I. T., Jain, A. K., Jones, S. D., Kato, E., Kennedy, D., Klein Goldewijk, K., Knauer, J., Korsbakken, J. I., Körtzinger, A., Landschützer, P., Lauvset, S. K., Lefèvre, N., Lienert, S., Liu, J., Marland, G., McGuire, P. C., Melton, J. R., Munro, D. R., Nabel, J. E. M. S., Nakaoka, S.-I., Niwa, Y., Ono, T., Pierrot, D., Poulter, B., Rehder, G., Resplandy, L., Robertson, E., Rödenbeck, C., Rosan, T. M., Schwinger, J., Schwingshackl, C., Séférian, R., Sutton, A. J., Sweeney, C., Tanhua, T., Tans, P. P., Tian, H., Tilbrook, B., Tubiello, F., van der Werf, G., Vuichard, N., Wada, C., Wanninkhof, R., Watson, A., Willis, D., Wiltshire, A. J., Yuan, W., Yue, C., Yue, X., Zaehle, S., and Zeng, J.: Global Carbon Budget 2021, Earth Syst. Sci. Data Discuss. [preprint], https://doi.org/10.5194/essd-2021-386, in review, 2021.

Lo Monaco, C., Metzl, N., Fin, J., Mignon, C., Cuet, P., Douville, E., Gehlen, M., Trang Chau, T.T., Tribollet, A., 2021. Distribution and long-term change of the sea surface carbonate system in the Mozambique Channel (1963-2019), *Deep-Sea Research Part II*, https://doi.org/10.1016/j.dsr2.2021.104936.

Ramanantsoa, J. D., Penven, P., Raj, R. P., Renault, L., Ponsoni, L., Ostrowski, M., et al.. Where and how the East Madagascar Current retroflection originates. *Journal of Geophysical Research: Oceans*, 126, e2020JC016203. https://doi.org/10.1029/2020JC016203, 2021

Figure R1: Top: Meridional section (Latitude/Depth) of zonal current (U in m.s$^{-1}$) observed from ADCP data collected in January 2020 in the South-Western Indian Ocean (OISO-30 cruise, see the track in Bottom). A strong westward current down to 600m is identified around 27-29°S. Figure produced with ODV (Schlitzer, 2013). Bottom: Map of monthly surface current for January 2020 in the South-Western Indian Ocean showing the retroflection of the East Madagascar Current here around 24°S (one of the forms of the EMC retroflection defined by Ramanantsoa et al 2021) and its complex meandering structure deflecting southward and recirculating northward around 54°E. Bottom Figure produced from https://resources.marine.copernicus.eu/ (MULTIOBS_GLO_PHY_REP_015_004) last access, 15-Dec-2021.

[Figure]

[Figure]

Figure R2: (a): Sea surface salinity (SSS, black circles) and zonal surface current (U cm/s, grey line, from ADCP data at 24m) in January 2020 along the OISO-30 cruise track around 54°E. (b): same as (a) for $fCO_2$ (µatm). The highest Chl-a concentration in the SEMB observed north of 27°S (Figure 3 in the Manuscript) in waters with lower salinity and low $fCO_2$ where the current was westward (U < 0, see figure R1). Westward current was also observed around 28-29°S in high salinity and high $fCO_2$ waters suggesting this is not a signal of an eddy.

[Figure]

---

## Author Comment (AC2)

Response to reviews of the paper: "The impact of the South-East Madagascar bloom on the oceanic CO2 sink" by Nicolas Metzl et al., Biogeosciences Discuss., https://doi.org/10.5194/bg-2021-283-RC1, 2021.

Reviewer 2, Ahmad Fehmi Dilmahamod

Reviewer comments in italic, our responses in red

We thanks reviewer 2, Ahmad Fehmi Dilmahamod for his enthusiastic review and suggestions.

*This study makes use of in-situ data from the OISO Project, among others, to understand the impact of the South-East Madagascar Bloom on the oceanic carbon sink. This is a first attempt to investigate how this large sporadic austral summer bloom can potentially contribute to the oceanic carbon sink. The physical drivers of this bloom have long been investigated with various possible mechanisms, although it seems to be coming down to a combination of a few factors which can contribute to the initiation of the bloom. On this note, the question of its impact on the oceanic carbon sink were previously raised but the lack of in-situ data hinders any research.*

*This very interesting study is well-structured and well-written. And with the right data, it provides new insights on the biogeochemical signature of this bloom. It clearly shows the difference between $fCO_2$ during a bloom and non-bloom (or low bloom) year, and that this difference is due to biological processes during the boom. And that it acts as a $CO_2$ sink (between -1.7 and -2.7 TgC/month).*

*Having said that, I am slightly less impressed with the reconstructed $fCO_2$ and air-sea $CO_2$ fluxes from Chau et al. (2021), which is still in review. I would expect that the Chau et al. (2021) is acepted before the current manuscript. I am also a bit puzzled because from Figures 11, S10 and S11, it seems that the impacts of the previous bloom years (1999, 2000, 2004, 2006, 2008, 2012-14) on the reconstructed $fCO_2$ and air-sea $CO_2$ flux are almost non-existent, whereas a significant drawdown is found for the 2020 one. I think that this variability from the CMEMS-LSCE-FFNNN is interesting, and deserve to be included and possibly explained in the text.*

Response: The reviewer is correct pointing to the low variability of reconstructed $CO_2$ fluxes in the model during other periods when the bloom was present but not as strong as observed in 2019-2020 (Figures 11, S10, S11). As can be seen on Figure S12 the low variability is also seen when fluxes are integrated over a relatively large domain (45-70E) and it is also seen in the $fCO_2$ data when available (Figure 2). However, data are not available for all years and we can only conclude that in 2000, 2006 and 2012 the observed $fCO_2$ was not low compared to the atmospheric concentrations as observed in 2020 (Figure 2). To highlight the impact of the bloom in 2019-2020 we thus focus the comparison on fluxes first for December (Figure 10) as well as for annual flux integrated in the region (Figure 11). We will add more information on the apparent low variability of fluxes for the full period investigated here (1997 to 2019).

*However, these do not take away the importance of in-situ data in this data-limited region and I am sure that the few comments can be easily addressed by the authors and that might help to improve this already good paper. Thus, I recommend this paper for publication, once the comments have been addressed, and the Chau et al. (2021) paper accepted.*

Response: The comment concerning the paper submitted by Chau et al (2021) was also suggested by Reviewer 1. The paper submitted by Chau et al (2021) was not reviewed at the time we prepared our paper. However, the reviews for their paper were posted on-line in late September and

the authors responded to the reviews last November 15th. At the time of this response (3/1/2022), the paper by Chau et al (2021) is not yet accepted but a revision is on the way.

Note that when we started this analysis (last year) we explored a first version of the same model described by Denvil-Sommer et al (2019) and Chau et la (2020) and preliminary results motivated the use of such a model for our purpose. The results of the first version of the model were also successfully used in another study focused in the Mozambique Channel (Lo Monaco et al, DSR 2021) and we were confident with this approach to explore the results of the model in the South-Western Indian sector. The model developed by Chau et al (2020, 2021) is an improvement of the first version by Denvil-Sommer et al (2019). Model results are under quality control of the European Copernicus Marine Environment Monitoring Service (CMEMS) and available for public use since 2019 (https://resources.marine.copernicus.eu/?option=com_csw&view=details&product_id=MULTIOBS_GLO_BIO_CARBON_SURFACE_REP_015_008, product DOI: https://doi.org/10.48670/moi-00047).

As the model has been updated, we thought it was more appropriate to use the new version of this model (Chau et al, 2021) extended to end-2019 that included the bloom anomaly well reproduced for December 2019 (Figures 10, 11).

Note also that results of the model by Chau et al (2021) have been used for global estimate of the ocean $CO_2$ sink in the last Global Carbon Budget (Friedlingstein et al, 2021; T. Chau is co-author and results presented at COP26 last November). Similarly, results of the CMEMS-LSCE-FFNN model (Denvil-Sommer et al 2019, Chau et al 2020) were also used in the previous GCB (Friedlingstein et al, 2019, 2020). Here we show the model is relevant for regional analysis, i.e. not only a view of the global ocean carbon sink as used in the Global Carbon Project. We hope that by the time our present paper is revised, the paper by Chau et al (2021) will be accepted.

***Minor Comment:***

*On lines 318-319, the authors mentioned the presence of a clear signal of the SEMC retroflection. A recent paper by Ramanantsoa et al. (2021) discussed the early-retroflection, retroflection and no retroflection of the SEMC, and the impacts of the early-retroflection on the SEMB. I recommend including this citation is the discussion.*

Response: Thank you, this is a good suggestion: we came across this article (Ramanantsoa et al, 2021) just after we submit our paper; these authors analyzed the complex circulation in this region for the period between 1987 and 2018 (before the bloom in 2019-2020) with a focus on the retroflection of the EMC. It is relevant to refer to this article when discussing the dynamics in this complex region and its potential impact on the SEMB. In this context, it is also relevant to note that in January 2020 (period of OISO-30 cruise) the EMC retroflection started before reaching the southern tip of the Madagascar Island, around 24°S (see figure R1) that corresponds to the so-called "Early Retroflection" as defined by Ramanantsoa et al (2021). We will add the information to the MS and the reference to Ramanantsoa et al (2021) will be added accordingly.

Note also that the reviewer's suggestions concerning either the "Eddy" (reviewer 1) or EMC retroflection (reviewer 2) leaded us to revisit the ADCP data in January-2020. We suggest to add a new figure to the Supp Mat (ADCP data in January 2020 and current field in this region) to describe the circulation during that period and it's potential link with SEMB. This figure is attached below (Figure R1).

;;;;;;;;;;; References added in this response (not listed in the manuscript):

Chau, T. T., Gehlen, M., and Chevallier, F., 2020: Global Ocean Surface Carbon Product MULTIOBS_GLO_BIO_CARBON_SURFACE_REP_015_008, E.U. Copernicus Marine Service

Information. "https://resources.marine.copernicus.eu/documents/QUID/CMEMS-MOB-QUID-015-008.pdf".

Denvil-Sommer, A., Gehlen, M., Vrac, M., and Mejia, C.: LSCE-FFNN-v1: a two-step neural network model for the reconstruction of surface ocean $p$CO$_2$ over the global ocean, Geosci. Model Dev., 12, 2091-2105, https://doi.org/10.5194/gmd-12-2091-2019,2019.

Friedlingstein, P., Jones, M. W., O'Sullivan, M., Andrew, R. M., Hauck, J., Peters, G. P., Peters, W., Pongratz, J., Sitch, S., Le Quéré, C., Bakker, D. C. E., Canadell, J. G., Ciais, P., Jackson, R. B., Anthoni, P., Barbero, L., Bastos, A., Bastrikov, V., Becker, M., Bopp, L., Buitenhuis, E., Chandra, N., Chevallier, F., Chini, L. P., Currie, K. I., Feely, R. A., Gehlen, M., Gilfillan, D., Gkritzalis, T., Goll, D. S., Gruber, N., Gutekunst, S., Harris, I., Haverd, V., Houghton, R. A., Hurtt, G., Ilyina, T., Jain, A. K., Joetzjer, E., Kaplan, J. O., Kato, E., Klein Goldewijk, K., Korsbakken, J. I., Landschützer, P., Lauvset, S. K., Lefèvre, N., Lenton, A., Lienert, S., Lombardozzi, D., Marland, G., McGuire, P. C., Melton, J. R., Metzl, N., Munro, D. R., Nabel, J. E. M. S., Nakaoka, S.-I., Neill, C., Omar, A. M., Ono, T., Peregon, A., Pierrot, D., Poulter, B., Rehder, G., Resplandy, L., Robertson, E., Rödenbeck, C., Séférian, R., Schwinger, J., Smith, N., Tans, P. P., Tian, H., Tilbrook, B., Tubiello, F. N., van der Werf, G. R., Wiltshire, A. J., and Zaehle, S.: Global Carbon Budget 2019, Earth Syst. Sci. Data, 11, 1783–1838, https://doi.org/10.5194/essd-11-1783-2019, 2019.

Friedlingstein, P., O'Sullivan, M., Jones, M. W., Andrew, R. M., Hauck, J., Olsen, A., Peters, G. P., Peters, W., Pongratz, J., Sitch, S., Le Quéré, C., Canadell, J. G., Ciais, P., Jackson, R. B., Alin, S., Aragão, L. E. O. C., Arneth, A., Arora, V., Bates, N. R., Becker, M., Benoit-Cattin, A., Bittig, H. C., Bopp, L., Bultan, S., Chandra, N., Chevallier, F., Chini, L. P., Evans, W., Florentie, L., Forster, P. M., Gasser, T., Gehlen, M., Gilfillan, D., Gkritzalis, T., Gregor, L., Gruber, N., Harris, I., Hartung, K., Haverd, V., Houghton, R. A., Ilyina, T., Jain, A. K., Joetzjer, E., Kadono, K., Kato, E., Kitidis, V., Korsbakken, J. I., Landschützer, P., Lefèvre, N., Lenton, A., Lienert, S., Liu, Z., Lombardozzi, D., Marland, G., Metzl, N., Munro, D. R., Nabel, J. E. M. S., Nakaoka, S.-I., Niwa, Y., O'Brien, K., Ono, T., Palmer, P. I., Pierrot, D., Poulter, B., Resplandy, L., Robertson, E., Rödenbeck, C., Schwinger, J., Séférian, R., Skjelvan, I., Smith, A. J. P., Sutton, A. J., Tanhua, T., Tans, P. P., Tian, H., Tilbrook, B., van der Werf, G., Vuichard, N., Walker, A. P., Wanninkhof, R., Watson, A. J., Willis, D., Wiltshire, A. J., Yuan, W., Yue, X., and Zaehle, S.: Global Carbon Budget 2020, Earth Syst. Sci. Data, 12, 3269–3340, https://doi.org/10.5194/essd-12-3269-2020, 2020.

Friedlingstein, P., Jones, M. W., O'Sullivan, M., Andrew, R. M., Bakker, D. C. E., Hauck, J., Le Quéré, C., Peters, G. P., Peters, W., Pongratz, J., Sitch, S., Canadell, J. G., Ciais, P., Jackson, R. B., Alin, S. R., Anthoni, P., Bates, N. R., Becker, M., Bellouin, N., Bopp, L., Chau, T. T. T., Chevallier, F., Chini, L. P., Cronin, M., Currie, K. I., Decharme, B., Djeutchouang, L., Dou, X., Evans, W., Feely, R. A., Feng, L., Gasser, T., Gilfillan, D., Gkritzalis, T., Grassi, G., Gregor, L., Gruber, N., Gürses, Ö., Harris, I., Houghton, R. A., Hurtt, G. C., Iida, Y., Ilyina, T., Luijkx, I. T., Jain, A. K., Jones, S. D., Kato, E., Kennedy, D., Klein Goldewijk, K., Knauer, J., Korsbakken, J. I., Körtzinger, A., Landschützer, P., Lauvset, S. K., Lefèvre, N., Lienert, S., Liu, J., Marland, G., McGuire, P. C., Melton, J. R., Munro, D. R., Nabel, J. E. M. S., Nakaoka, S.-I., Niwa, Y., Ono, T., Pierrot, D., Poulter, B., Rehder, G., Resplandy, L., Robertson, E., Rödenbeck, C., Rosan, T. M., Schwinger, J., Schwingshackl, C., Séférian, R., Sutton, A. J., Sweeney, C., Tanhua, T., Tans, P. P., Tian, H., Tilbrook, B., Tubiello, F., van der Werf, G., Vuichard, N., Wada, C., Wanninkhof, R., Watson, A., Willis, D., Wiltshire, A. J., Yuan, W., Yue, C., Yue, X., Zaehle, S., and Zeng, J.: Global Carbon Budget 2021, Earth Syst. Sci. Data Discuss. [preprint], https://doi.org/10.5194/essd-2021-386, in review, 2021.

Lo Monaco, C., Metzl, N., Fin, J., Mignon, C., Cuet, P., Douville, E., Gehlen, M., Trang Chau, T.T., Tribollet, A., 2021. Distribution and long-term change of the sea surface carbonate system in the

Mozambique Channel (1963-2019), *Deep-Sea Research Part II*, https://doi.org/10.1016/j.dsr2.2021.104936.

Ramanantsoa, J. D., Penven, P., Raj, R. P., Renault, L., Ponsoni, L., Ostrowski, M., et al.. Where and how the East Madagascar Current retroflection originates. *Journal of Geophysical Research: Oceans*, 126, e2020JC016203. https://doi.org/10.1029/2020JC016203, 2021

Figure R1: Top: Meridional section (Latitude/Depth) of zonal current (U in m.s$^{-1}$) observed from ADCP data collected in January 2020 in the South-Western Indian Ocean (OISO-30 cruise, see the track in Bottom). A strong westward current down to 600m is identified around 27-29°S. Figure produced with ODV (Schlitzer, 2013). Bottom: Map of monthly surface current for January 2020 in the South-Western Indian Ocean showing the retroflection of the East Madagascar Current here around 24°S (one of the forms of the EMC retroflection defined by Ramanantsoa et al 2021) and its complex meandering structure deflecting southward and recirculating northward around 54°E. Bottom Figure produced from https://resources.marine.copernicus.eu/ (MULTIOBS_GLO_PHY_REP_015_004) last access, 15-Dec-2021.

[Figure]

[Figure]

---

## Author Response (AR1)

Authors response to the Associate Editor report

Title: The impact of the South-East Madagascar bloom on the oceanic CO2 sink
Author(s): Nicolas Metzl et al.
MS No.: bg-2021-283
MS type: Research article
Iteration: Minor revision

;;;;;;;;;;;; Associate Editor report

Dear authors,

many thanks for addressing the referee comments. Both referees agree that the study is well suited for publication in Biogeosciences, however, both referees equally raise concern regarding parts of this study being linked to a paper currently in review.

As explained in the referee response document, the Chau et al (in review) study comprises an update of a past study (Denvil-Sommer et al 2019), which I am well familiar with and which has appeared in the peer-reviewed literature. Therefore, I am not concerned regarding this update not being tight to a published manuscript yet. I would however suggest that the authors make it clear in this manuscript (for the readers of this study) that they present an improvement of a previously peer-reviewed method. I believe that this would increase the readers confidence.

Additionally, as BG is striving to improve the accessibility of colour figures for readers with colour vision deficiencies, please have a look at the BG figure guidelines (https://www.biogeosciences.net/submission.html#figurestables).

Overall, based on the referee comments, I believe that minor revisions are required before the manuscript can be considered for publication.

Best regards
Peter Landschützer

;;;;;;;;;;; Our response:

Dear Editor, dear Peter,

Thanks you for your positive comments on our manuscript.

The manuscript has been revised with main revisions listed below:

In Section 3, we have specified that the model from Chau et al, (2022 now accepted in BG) is an updated version of the previous CMEMS-FFNN model by Denvil-Sommer et al (2019), now added in references.

We have added reference to a very recent paper by Ramanantsoa et al (2021) as suggested by reviewer 2. This is article is now referenced in several places in the manuscript when describing the current fields and specifically when identifying the retroflection of the SEMC.

As suggested by reviewer 2, we also added a note in section 5.3 concerning the relatively low variability of the annual $CO_2$ sink estimated by the CMEMS-FFNN model during other blooms compared to the strong bloom that occurred in 2019-2020.

As we suggested in our reply to the reviews, we decided to add a new figure in Supp Mat (now Figure S4, ADCP data recorded during the OISO-30 cruise and a map of the currents in January 2020, not presented in the submitted manuscript). The Figure numbers in the Supp. Mat and the main text have been revised accordingly.

As suggested by reviewer 1 we added gridded of Latitude/longitude in all maps.

We have also changed the colors for all figures following the Color Blindness Simulator as suggested by BG (revised figures in the main MS and the Supplement). We hope the new colors will be correct for all readers (including colorblind readers) and the main results are clear for everyone.

The revised paper and the Supplement are submitted as well as a version with track-changes for the text (few lines revised that should be clearly identified).

We hope these changes are completed and the manuscript suitable for publication.

Thank you again to handle this manuscript and for your positive decision

Sincerely,

Nicolas Metzl on behalf of all co-authors